# SO$_2$ and NH$_3$ emissions enhance organosulfur compounds and fine particles formation from the photooxidation of a typical aromatic hydrocarbon

Zhaomin Yang[1], Li Xu[1], Narcisse T. Tsona[1], Jianlong Li[1], Xin Luo[2], Lin Du[1]

[1]Environment Research Institute, Shandong University, Qingdao, 266237, China
[2]Technology Center of Qingdao Customs, Qingdao, 266003, China
*Correspondence to*: Lin Du (lindu@sdu.edu.cn)

**Abstract.** Aromatic hydrocarbons can dominate the volatile organic compounds budget in the urban atmosphere. Among them, 1,2,4-trimethylbenzene (TMB), mainly emitted from solvent use, is one of the most important secondary organic aerosols (SOA) precursors. Although atmospheric SO$_2$ and NH$_3$ levels can affect secondary aerosol formation, the influenced extent of their impact and their detailed driving mechanisms are not well understood. The focus of the present study is to examine

the chemical compositions and formation mechanisms of SOA from TMB photooxidation influenced by SO$_2$ and/or NH$_3$. Here, we showed that SO$_2$ emission could considerably enhance aerosol particle formation due to SO$_2$-induced sulfate generation and acid-catalyzed heterogeneous reaction. Orbitrap mass spectrometry (MS) measurements revealed the generation of not only typical TMB products but also hitherto unidentified organosulfates (OSs) in SO$_2$-added experiments.

The OSs designated as unknown origin in earlier field measurements were also detected in TMB SOA, indicating that atmospheric OSs might be also originated from TMB photooxidation. For NH$_3$-involved experiments, results demonstrated a positive correlation between NH$_3$ levels and particle volume as well as number concentrations. The effects of NH$_3$ on SOA composition was slight under SO$_2$-free conditions but stronger in the presence of SO$_2$. A series of multifunctional

products with carbonyl, alcohols, and nitrate functional groups were tentatively characterized in NH$_3$-involved experiments based on infrared spectra and MS analysis. Plausible formation pathways were proposed for detected products in the particle-phase. The volatility distributions of products, estimated using parameterization methods, suggested that the detected products gradually condense onto the nucleation particles to contribute to aerosol formation and growth. Our results suggest that

strict control of SO$_2$ and NH$_3$ emissions might remarkably reduce organosulfates and secondary aerosol burden in the atmosphere. Updating the aromatic oxidation mechanism in models could result in more accurate treatment of particles formation for urban regions with considerable SO$_2$, NH$_3$, and aromatics emissions.

## 1 Introduction

Secondary organic and inorganic aerosols have been observed to account for a considerable fraction of fine particulate matter (aerosol particles $\leq 2.5$ μm in aerodynamic diameter, $PM_{2.5}$) during $PM_{2.5}$ pollution events, which had frequently occurred and lasted for days or even weeks in China during the last decade (Huang et al., 2014; Guo et al., 2014). These particles can directly and indirectly impact regional and global climate (Kanakidou et al., 2005), air quality (Zhang et al., 2015), and human health (Lelieveld et al., 2015).

Secondary organic aerosols (SOA) arise predominantly from the oxidation of volatile organic compounds (VOCs) in the atmosphere. Early atmospheric models underestimated the measured SOA mass concentrations in field studies by 1–2 orders of magnitude (Volkamer et al., 2006; Heald et al., 2010). Although recent efforts such as updating missing SOA precursors, accounting unknown processes of gas-to-particle conversion, and improving emission inventories have narrowed the observed gap between the modeled and measured SOA mass, uncertainties still exist in organic aerosol estimates (Shrivastava et al., 2011; Cheng et al., 2021). Inorganic perturbations on SOA formation (Shrivastava et al., 2017) are partly responsible for these uncertainties, and they include the addition of mineral particles (Yu and Jang, 2019), nitrogen oxide ($NO_x$) (Zhao et al., 2018), ammonia ($NH_3$) (Hao et al., 2020), and sulfur dioxide ($SO_2$) (Yang et al., 2020), which can engage in the gas- or particle-phase chemistry and subsequently influence SOA formation and growth (Friedman et al., 2016; Ng et al., 2007; Na et al., 2006). $NO_x$ effects on particle formation are generally known to be pronounced. High levels of $SO_2$, $NH_3$, and VOCs have been reported in certain regions such as Guangzhou (Zou et al., 2015), Beijing (Meng et al., 2020), Handan (Li et al., 2017) in China. During haze pollution episodes, Li et al. (2017) observed that $SO_2$ levels can be up to 200 ppb in Handan, China. A recent study also showed significant increasing $NH_3$ levels in the atmosphere over the United States and the European Union (Warner et al., 2017). However, less focus has been placed on the $SO_2$ and $NH_3$ perturbations on SOA formation and chemical composition. Aerosol particles contain a multitude of compounds with different physicochemical properties. Previous laboratory studies examined the photooxidation of cyclohexene, fuel, and 1,3,5-trimethylbenzene in the presence of $SO_2$ and reported organosulfates (OSs) formation (Yang et al., 2020; Liu et al., 2017; Blair et al., 2017). The atmospheric oxidation of $SO_2$ can generate sulfuric acid that is critical for the increase of particle acidity. $SO_2$-induced acidic sulfate plays an active role in the production of OSs, which have been recognized as significant SOA tracers describing the enhancement in SOA by $SO_2$ emission (Xu et al., 2015).

OSs are ubiquitous in ambient aerosol particles and they are estimated to account for 3–30% of the organic mass in fine aerosol particles (Surratt et al., 2008; Tolocka and Turpin, 2012). The presence of OSs could alter aerosol morphology (Riva et al., 2019), viscosity (Riva et al., 2019; Zhang et al.,

2019), particle acidity (Riva et al., 2019), phase state (Zhang et al., 2019), hygroscopicity (Estillore et al., 2016; Hansen et al., 2015), and optical properties (Fleming et al., 2019), thereby resulting in large climate effects. A large number of OSs have previously been observed in field measurements

but only a few biogenic VOCs can be clearly designated as OSs precursors (Wang et al., 2019b; Shalamzari et al., 2014). Recent field studies reported that some unidentified OSs with $C_2$–$C_{25}$ skeletons may not be originated from biogenic VOCs and suggested that anthropogenic VOCs may contribute to these OSs formation (Wang et al., 2016; Blair et al., 2017). In addition, Ma et al. (2014) demonstrated that OSs derived from aromatic hydrocarbons contribute to up to 67% of the total OSs

mass in Shanghai, China, highlighting the potentially significant role of anthropogenic aromatics in organosulfate formation. While several studies have shown that $SO_2$ emissions have implications for the SOA burden and OSs formation, detailed characterizations of OSs formation from anthropogenic monocyclic aromatic photooxidation are poorly performed.

$NH_3$ is the most abundant form of reduced nitrogen and it is ubiquitous in the ambient environment. $NH_3$ levels increased substantially in recent years and are estimated to continue to increase in the future (Warner et al., 2017). It is established that the increased $NH_3$ emissions could reduce the effectiveness of $PM_{2.5}$ control by controlling $SO_2$ and $NO_x$ (Wu et al., 2016; Fu et al., 2017). However, the effects of $NH_3$ on the formation of aerosol particles have not been well understood.

$NH_3$ has a promoting effect on the formation of new particles (Wang et al., 2020a) where low volatile organic compounds could condense to form SOA in the atmosphere. A previous chamber study reported that the addition of $NH_3$ could lead to the enhancement in the volume and number concentrations of SOA from $\alpha$-pinene ozone system (Na et al., 2007). Another study by Babar et al. (2017) utilized newly developed flow reactor and confirmed that the presence of $NH_3$ can enhance

SOA formation from both ozonolysis and photooxidation of $\alpha$-pinene. Besides $\alpha$-pinene, the promoting effects of $NH_3$ on particle formation were also discovered in the photooxidation of aromatics (Chu et al., 2016) and vehicle exhaust (Chen et al., 2019). In contrast, addition of $NH_3$ decreased aerosol particle formation from the reaction of styrene with ozone owing to the decomposition of products by $NH_3$ nucleophilic attack (Na et al., 2006). Laboratory evidence

suggests that $NH_3$ can influence SOA composition via the neutralization of organic acids (Hao et al., 2020) and via the $NH_3$ uptake by carbonyl-containing compounds (Flores et al., 2014). The reaction of organic compounds in particle-phase with $NH_3$ can decrease gaseous $NH_3$ concentrations and can enhance the formation of nitrogen-containing organic compounds (Liu et al., 2015b), which are a class of brown carbon and could modify SOA optical properties. The neglect of $NH_3$ effects on SOA

formation might increase the model-measurement disagreement in SOA mass and can lead to an overprediction of $NH_3$ concentration in the gas-phase, especially in a complex urban environment. Consequently, it is necessary to explicitly explore the influence of $NH_3$ on aerosol particles formation.

The complex mixture of ozone and fine particles is an emerging environmental problem affecting regional and urban air quality in China (Song et al., 2017), and investigating the chemistry of aromatic hydrocarbons has become greatly important for ozone and $PM_{2.5}$ control because aromatics have high ozone- and SOA-forming potential (Chu et al., 2020). Aromatic hydrocarbons comprise a substantial fraction of the total VOCs at urban locations and even in rural areas (Guo et al., 2006;

Ran et al., 2009), and evidence shows that global SOA formation from aromatic hydrocarbons lies in the range of 2 to 12 Tg $yr^{-1}$ (Henze et al., 2008). 1,2,4-Trimethylbenene (TMB) is a small monocyclic aromatic emitted primarily from industrial solvent evaporation (Mo et al., 2021). In the troposphere, TMB is mainly oxidized via hydroxyl radical (OH), producing multigenerational oxidized compounds that can contribute to SOA formation and growth (Zaytsev et al., 2019; Mehra

et al., 2020). The OH oxidation of TMB can also generate small dicarbonyls glyoxal and methylglyoxal (Zaytsev et al., 2019), which are significant precursors for light-absorbing SOA formation. In addition, the Master Chemical Mechanism (MCM, http://mcm.leeds.ac.uk/MCMv3.2/, last access: 23 February 2021) is a near-explicit chemical mechanism that can describe, in detail, the tropospheric degradation of TMB. A recent study reported that identified autoxidation pathways

during OH oxidation of TMB are not included in the current MCM and the detected TMB products are more diverse than the products shown in MCM (Wang et al., 2020b). The updates for the OH-initiated oxidation mechanism of TMB can be achieved only when the rate constants, branching ratios and product distributions can be explicitly obtained. However, TMB photooxidation is highly complex and sensitive to environmental conditions. To better understand TMB SOA formation and

growth, investigating chemical processes of TMB photooxidation with inorganic perturbation is required.

The mechanisms leading to secondary aerosol formation in the urban environment remain highly elusive and controversial, particularly for the processes related to changes in secondary aerosol mass

and chemical composition. Recent studies have suggested that inorganic pollution emissions could perturb SOA formation, yet very little is known about the $SO_2$ and $NH_3$ effects on SOA formation. Given the ubiquity of $SO_2$, $NH_3$, and TMB in the atmosphere, a key goal of this work is to determine the detailed chemical compositions and formation mechanisms of secondary aerosol from TMB photooxidation with $SO_2$ and/or $NH_3$. We investigated the effects of $SO_2$ and $NH_3$ on the growth of

particles from TMB photooxidation for the first time and discussed the role of inorganic species in TMB chemistry. The chemical composition of TMB SOA was rigorously characterized based on laboratory measurements. We also revealed some hitherto unidentified organosulfates and tentatively proposed relevant formation pathways of products.

## 2 Experimental methods

### 2.1 Particle generation

Aerosol particles were produced from TMB photooxidation in the presence of $NO_x$ in a new indoor smog chamber, which consists of a 1.1 $m^3$ Teflon reactor (0.6 mm Teflon film) housed in a temperature-controlled room. For photooxidation, a panel of black light lamps (F40BLB, GE) were used to provide ultraviolet (UV) irradiation centered at 365 nm. Before each run, the chamber was continually purged with dry and purified air prepared by zero air supply (Model 111, Thermo Scientific, USA) and simultaneously irradiated with UV lights until the concentrations of background contaminants (i.e., NO, $NO_2$, $SO_2$, and $O_3$) were lower than 1 ppb and the particle number concentration was below 5 $cm^{-3}$.

The TMB photooxidation experiments were carried out by the following steps. First, a known volume of TMB liquid (98%, Aladdin) was transported into the chamber through a heated (80 °C) Teflon tube carried by a flow of zero air. Second, according to experimental design, different quantities of NO (504 ppm in $N_2$, Qingdao Yuyan Gas Company, China), $SO_2$ (1013 ppm in $N_2$, Qingdao Yuyan Gas Company, China), and $NH_3$ (497 ppm in $N_2$, Qingdao Yuyan Gas Company, China) were introduced into the chamber from corresponding high-pressure cylinders using calibrated mass flow controllers (D07-7, Beijing Sevenstar Electronics Co., Ltd, China). Note that before the injection of $NH_3$, the inlet tubes were flushed with $NH_3$ flow for 30 min to minimize the adsorption losses of $NH_3$ in the tubes. After all species in the chamber were well mixed (initial concentrations of TMB, NO, and/or $SO_2$ were constant), black light lamps were turned on, marking the beginning of photooxidation experiments. The chamber was operated in batch mode with a reaction time of approximately 300–360 min. Seed particles were not introduced into the chamber over the course of particle formation experiments. Temperature and relative humidity (RH) inside the chamber were (299 ± 4) K and (25 ± 1) %, respectively. Detailed experimental conditions and results for each experiment are provided in Table 1. Twelve experiments were conducted under four different scenarios. In the first set of experiments (Exps. 1–4, Table1), $SO_2$ levels in the chamber varied from 0 to 200 ppb while the initial ratio of [TMB] to [$NO_x$] was kept higher than 10 ppbC $ppb^{-1}$ (low-$NO_x$ condition). The second set of experiments (Exps. 5–8, Table1) were performed under high-$NO_x$ condition ([TMB]$_0$/[$NO_x$]$_0$ < 10 ppbC $ppb^{-1}$) with $SO_2$ concentration being the only variable (ranged from 0 to 228 ppb). The third part of experiments (Exps. 9–10, Table1) consisted of an irradiation of TMB, $NO_x$, and $NH_3$, while the subsequent part of experiments (Exps. 11–12, Table1) consisted of an irradiation of TMB, $NO_x$, $SO_2$, and $NH_3$.

Different physicochemical parameters were measured over the course of photooxidation experiments. A digital thermo-hydrometer (Model 645, Testo AG, Lenzkirch, Germany) was used to measure temperature and RH inside the chamber. The concentrations of NO and $NO_x$ were

measured with a NO-NO$_2$-NO$_x$ analyzer (model 42i, Thermo scientific, USA), while a Thermo scientific model 43i-TLE pulsed fluorescence SO$_2$ analyzer was used to measure SO$_2$ levels throughout the experiments. The O$_3$ level was monitored with a Thermo scientific model 49i O$_3$ analyzer. The initial concentration of NH$_3$ was calculated based on the introduced amount of NH$_3$

and the reactor volume. The decay of TMB was measured by a gas chromatography (GC, 7890B, Agilent Technologies, USA) equipped with a DB-624 column (30 m × 0.32 mm, 1.8 $\mu$m film thickness, Agilent Technologies, USA) and a flame ionization detector (FID). The GC temperature was programmed to increase from 80 to 200 °C at 20 °C min$^{-1}$ rate. Particle size distributions and volume concentrations in all experiments were recorded in situ using a scanning mobility particle

sizer (SMPS), which consists of a long differential mobility analyzer (long-DMA, Model 3082, TSI, USA) and a condensation particle counter (CPC, Model 3776, TSI, USA). The long-DMA was available for measuring the particle size distribution in the range of 13.8–723.4 nm while smaller particles between 4.5–162.5 nm were measured with a nano differential mobility analyzer (nano-DMA, Model 3085, TSI, USA). The measured volume concentration in each experiment was

converted into particle mass concentration with an estimated particle density of 1.4 g cm$^{-3}$.

Table 1. Experimental conditions and results for the TMB photooxidation experiments.

| Exp. | [TMB]$_0$ (ppb) | [TMB] consumed ($\mu$g m$^{-3}$) | [TMB]$_0$/[NO$_x$]$_0$ (ppbC ppb$^{-1}$) | [NO$_x$]$_0$ (ppb) | [SO$_2$]$_0$ (ppb) | [NH$_3$]$_0$ (ppb) | [OH] $\times 10^{-6}$ (molecules cm$^{-3}$) [a] | RH (%) | T (K) | Surface area concentration $\times 10^{-3}$ ($\mu$m$^2$ cm$^{-3}$) [b] | N$_{max}\times 10^{-5}$ (cm$^{-3}$) [c] | SOA mass ($\mu$g m$^{-3}$) [d] | SOA yield (%) [e] |
|---|---|---|---|---|---|---|---|---|---|---|---|---|---|
| 1 | 374 | 1404 | 18.9 | 178 | - | - | 2.52 | 25 | 295 | 1.08 | 0.27 | 52.6 | 3.8 ± 0.4 |
| 2 | 350 | 1385 | 17.3 | 182 | 59 | - | 2.33 | 24 | 296 | 2.33 | 1.12 | 97.8 | 8.2 ± 0.7 |
| 3 | 368 | 1303 | 17.3 | 191 | 107 | - | 2.29 | 25 | 296 | 2.82 | 1.13 | 164.8 | 12.6 ± 1.3 |
| 4 | 220 | 766 | 10.0 | 199 | 200 | - | 3.00 | 25 | 295 | 3.45 | 1.40 | 175.6 | 17.6 ± 2.0 |
| 5 | 393 | 1693 | 7.6 | 465 | - | - | 3.62 | 26 | 298 | 1.21 | 0.29 | 59.4 | 3.5 ± 0.4 |
| 6 | 346 | 1599 | 6.8 | 457 | 68 | - | 3.44 | 24 | 295 | 2.65 | 0.93 | 120.7 | 7.9 ± 0.7 |
| 7 | 260 | 1069 | 5.1 | 457 | 114 | - | 3.12 | 26 | 298 | 2.06 | 0.98 | 78.7 | 8.0 ± 0.7 |
| 8 | 379 | 1534 | 7.4 | 464 | 228 | - | 3.06 | 24 | 297 | 3.76 | 1.10 | 186.9 | 12.2 ± 1.2 |
| 9 | 454 | 1880 | 8.6 | 475 | - | 100 | 3.42 | 26 | 303 | 1.58 | 0.45 | 70.6 | 3.8 ± 0.4 |
| 10 | 464 | 1946 | 9.1 | 457 | - | 200 | 3.37 | 25 | 299 | 2.13 | 0.60 | 99.3 | 5.1 ± 0.5 |
| 11 | 425 | 1972 | 8.6 | 447 | 227 | 100 | 3.51 | 24 | 299 | 4.45 | 1.57 | 209.6 | 12.9 ± 1.3 |
| 12 | 450 | 1974 | 8.9 | 455 | 234 | 200 | 3.32 | 26 | 303 | 4.80 | 2.00 | 244.9 | 13.7 ± 1.4 |

[a] The average OH concentration was determined from the measured TMB decay. [b] The particle surface area concentration measured by SMPS at 300 min of each experiment. [c] Aerosol particle maximum number concentration. [d] SOA mass concentrations have been wall-loss corrected. For SOA mass calculation, the inorganic mass concentration has been subtracted from the particle mass concentration. [e] Errors in SOA yield were calculated from error propagation using the sum of the uncertainties in TMB data and the systematic error of SMPS.


## 2.2 Particle collection and analysis

### 2.2.1 Attenuated total reflectance-Fourier transform infrared spectroscopy analysis

Following 300–360 min of reaction, aerosol samples were collected onto aluminum foils (25 mm, Jowin Technology Co. Ltd.) by a low-pressure impactor (DLPI+, DeKati Ltd, Finland) and were stored at -20 °C thereafter until analysis to reduce evaporative losses of aerosol. The chemical functional groups of aerosols were characterized by an attenuated total reflectance-Fourier transform infrared (ATR-FTIR) spectrophotometer (Vertex 70, Bruker, Germany) with mercury

cadmium telluride detector. The ATR-FTIR spectra of aerosol particles in each run were recorded by averaging 64 scans from 4000–600 $cm^{-1}$ with a resolution of 4 $cm^{-1}$. Prior to each measurement with ATR-FTIR, the surface of the diamond crystal was thoroughly cleaned with ethanol and ultrapure water to rule out interferences of other sources of contamination. The ATR-FTIR spectra of blank aluminum foils were also acquired to confirm the absence of IR absorption by the aluminum

foil on which aerosols are collected.

### 2.2.2 Ion chromatography analysis

Following the ATR-FTIR measurements, aerosol samples were extracted in 3 mL of ultrapure water (Milli-Q water, 18.2 MΩ) under sonication in an ice bath for 30 min. The extracted samples were filtered through polyethersulfone syringe filters (0.22 $\mu$m pore size) and subsequently analyzed for

their ionic concentrations using an ion chromatography (Dionex ICS-600, Thermo Fisher Scientific, USA) with electrical conductivity detection. A Dionex IonPac$^{TM}$ AS19 column (4 × 250 mm) connected with AG19 guard column (4 × 50 mm, Dionex Ionpac) was used to separate anions. An aqueous solution of 20 mM potassium hydroxide (KOH) prepared by reagent-free controller (Dionex, Thermo Scientific, USA) was used as anion eluent. Cation analysis was carried out with

the pair of CG12A guard column (4 × 50 mm, Dionex Ionpac) and analytical column (4 × 250 mm, CS12A, Dionex IonPac$^{TM}$) and an isocratic 20 mM methanesulfonic acid ($CH_4O_3S$). The same volume of extract was injected into the ion chromatograph by a six-way valve mounted with a loop of 250 $\mu$L. The elution flow rates of KOH and $CH_4O_3S$ were both set to 1 L $min^{-1}$.

### 2.2.3 Ultra-high-performance liquid chromatography high resolution mass spectrometry

**analysis**

Laboratory-generated aerosols were also collected on 47 mm polytetrafluoroethylene (PTFE) filters (0.22 $\mu$m pore size, Tianjin Jinteng Experimental Equipment, China) using a stainless steel inline filter holder (Sartorius 16254, Sartorius Stedim Biotech GmbH, Germany) with a flow rate of 10 L $min^{-1}$. The collected samples were wrapped in foil and stored in the freezer (-20 °C) until mass

spectrometry analysis. Filter samples were extracted twice with 5 mL of high-purity methanol

(Optima® LC-MS grade, Fisher Scientific) under sonication in ice for 30 min. The extracts were mixed, filtered with a 0.2 $\mu$m pore size PTFE syringe filter (Millipore), and concentrated to near dryness under a gentle stream of high-purity nitrogen. The concentrated samples were reconstituted with ultrapure water (Milli-Q water, 18.2 M$\Omega$) and methanol (Optima® LC-MS grade, Fisher Scientific) with a volume ratio of 50:50. Control mass spectrometry measurements of solvent and extracts from blank PTFE filters were preformed to remove the interferences of solvent and handling protocols. The chemical compositions of aerosols were characterized using an ultra-high-performance liquid chromatography (UPLC, Ultimate 3000, Thermo scientific, USA) coupled to a Q-Exactive Focus Hybrid Quadrupole-Orbitrap mass spectrometry (MS, Thermo Scientific, USA) with electrospray ionization (ESI). The ESI source was operated in both positive (+) and negative (-) ionization mode. Product molecules could be detected as $[M + H]^+$ in the positive ion mode while products could be ionized via deprotonation and were detected as $[M - H]^-$ in the negative ion mode. The following parameters were set for the optimal operation of LC/ESI-MS: spray voltage (+), 3.5 kV; spray voltage (-), -3.0 kV; S-lens RF level (+), 50 V; S-lens RF level (-), 50 V; capillary temperature, 320 °C; sheath gas (nitrogen) pressure, $2.76 \times 10^5$ Pa; auxiliary gas (nitrogen) flow, 3.33 L min$^{-1}$. MS spectra were recorded in the range of $m/z$ 50 to 750 in full MS scan with a mass resolving power of 70000 (FWHM at $m/z$ 200). The full MS scan was followed by data-dependent MS/MS (dd-MS$^2$) scans using stepped collision energies of 20, 40, and 60 eV via high-energy collisional dissociation. The resolution was 17500 and an isolation width of 2 $m/z$ units was applied for the dd-MS$^2$ scan. The other parameters for MS$^2$ experiments were as follow: AGC target, $2 \times 10^5$; maximum IT, 50 ms; loop count, 3; minimum AGC target, $1 \times 10^5$; apex trigger, 2–6 s; dynamic exclusion, 6 s. The MS instrument was calibrated every five days with standard calibration solutions provided by the manufacturer. The separation of analytes was carried out on an Atlantis T3 C18 column (100 Å, 3 mm particle size, 2.1 mm × 150 mm, Waters, USA) at 35 °C. The mobile phases consisted of (A) 0.1% formic acid (Optima® LC-MS grade, Fisher Scientific) in ultra-pure water (Milli-Q water, 18.2 M$\Omega$) and (B) 0.1% formic acid in methanol (Optima® LC-MS grade, Fisher Scientific). The injected volume of samples was 2 $\mu$L in this study. Samplers were eluted using a 60-min gradient elution program with a flow rate of 200 $\mu$L min$^{-1}$: initially set to 3% B over the first 3 min, the concentration of eluent B was increased linearly to 50% in 22 min, from 50% to 90% from 25 min to 43 min, then it was decreased from 90% to 3% from 43 to 48 min, and finally kept at 3% for 12 min. The chemical formulas of observed ions were proposed based on reaction pathways, chemical consideration, and measured $m/z$ value with a mass tolerance of ± 5 ppm. All data were recorded and processed using Xcalibur V4.2.47 software package.

**2.3 Wall losses of vapors and particles**

The SOA yield, usually used to quantify the propensity of a parent hydrocarbon to form SOA, could be determined as the ratio of the generated particle mass to the amount of consumed parent hydrocarbon. The particle and vapor wall depositions in chambers can lead to the underestimation

of the SOA yield. In order to determine the particles wall loss rates, we carried out independent wall loss experiments using ammonium sulfate particles. An aqueous solution of ammonium sulfate was fed to a constant output atomizer (Model 3706, TSI, USA) to produce droplets, which passed simultaneously through a silica gel diffusion dryer to introduce dry particles into the chamber. The size distributions of ammonium sulfate particles were measured by SMPS for 480 min. Almost identical humidity ((25 ± 1) %) condition was achieved among each experiment. The wall losses of particles are size-dependent and, thus, we used a size-dependent particle wall-loss correction approach, which is described in detail in the Supplement. The size-dependent loss rate ($k$) of ammonium sulfate particles could be expressed as $k = 5.5 \times 10^{-6} \times d_p^{1.05} + 0.18 \times d_p^{-1.19}$ and was applied to correct the aerosol particle concentrations. The wall loss rates of NO, $NO_2$, $SO_2$, and TMB were determined to be $2.0 \times 10^{-6}$, $3.9 \times 10^{-6}$, $4.0 \times 10^{-7}$, and $2.3 \times 10^{-6}$ $s^{-1}$, respectively, indicating that the wall losses of these species were negligible over the course of the experiment. However, gas-phase species that could deposit to the chamber walls include not just the parent hydrocarbon, in this case TMB, but also the oxidation products, which in general are not all totally monitored and characterized. It is difficult to correct directly and accurately for the impact of vapor wall losses on the SOA yield. Therefore, the SOA yield in this work is only the lower limit. Here, the underestimation of SOA yields due to vapor wall losses was determined by the method in the study of Zhang et al. (2014). In brief, the effect of vapor wall losses on SOA yield significantly arose from the competition between vapors condensation onto aerosol particles versus vapor depositions to chamber walls. The extent to which vapor wall depositions influence the SOA yield could be estimated by the ratio of the average timescale of gas-particle partitioning during the photooxidation experiments to the timescale of vapor wall deposition. The evaluated results suggested that the SOA yield could be underestimated by a factor of 1.8 to 8.4 without accounting for vapor losses.

## 3 Results and discussion

### 3.1 Effects of $SO_2$

### 3.1.1 Particle formation and growth in $SO_2$-added photooxidation

To evaluate the impacts of $SO_2$ on aerosol formation and growth from TMB photooxidation, a series of experiments were conducted with various initial $SO_2$ levels under both low- and high-$NO_x$ conditions. Evolutions of the number distributions of secondary aerosols with particle size within 4.5–162.5 nm are provided in Fig. 1. There was no new secondary aerosol formation in the beginning of photooxidation. After a period of time, particles were burst produced and the number concentration of particle increased rapidly. At the same time, the particles continuously grew via condensation and coagulation mechanisms, consistent with a previous study (Jorga et al., 2020). After 300 min UV irradiation without $SO_2$ introduction, the total maximum number concentration of aerosol particles was only $2.7 \times 10^4$ $cm^{-3}$ and $2.9 \times 10^4$ $cm^{-3}$ under low-$NO_x$ (Exp. 1) and high-

NO$_x$ (Exp. 5) experiments, respectively. Interestingly, the particle maximum number concentration considerably increased with increasing SO$_2$ levels, regardless of low- or high-NO$_x$ conditions (Table

1). As shown in Fig. 1, although the unimodal particle size distribution in the presence of SO$_2$ was similar to that in its absence, particles with mobility diameter ranging from 10 to 80 nm, especially, dominated particle number concentrations with SO$_2$ addition. The time series of total ultrafine particles (diameter < 100 nm) number concentration are shown in Fig. S2, where it is seen that SO$_2$ considerably enhanced ultrafine particle formation. Ultrafine particles are more harmful to human

than larger particles because they can more easily penetrate deep into the lungs and blood circulation (Terzano et al., 2010). Our results indicate that SO$_2$ concentration is a key parameter for ultrafine particle formation. It has been suggested that reducing the number concentration of ultrafine particles can decrease mortality, highlighting again the need to continue to implement strict SO$_2$ emission standards (Fuzzi et al., 2015).


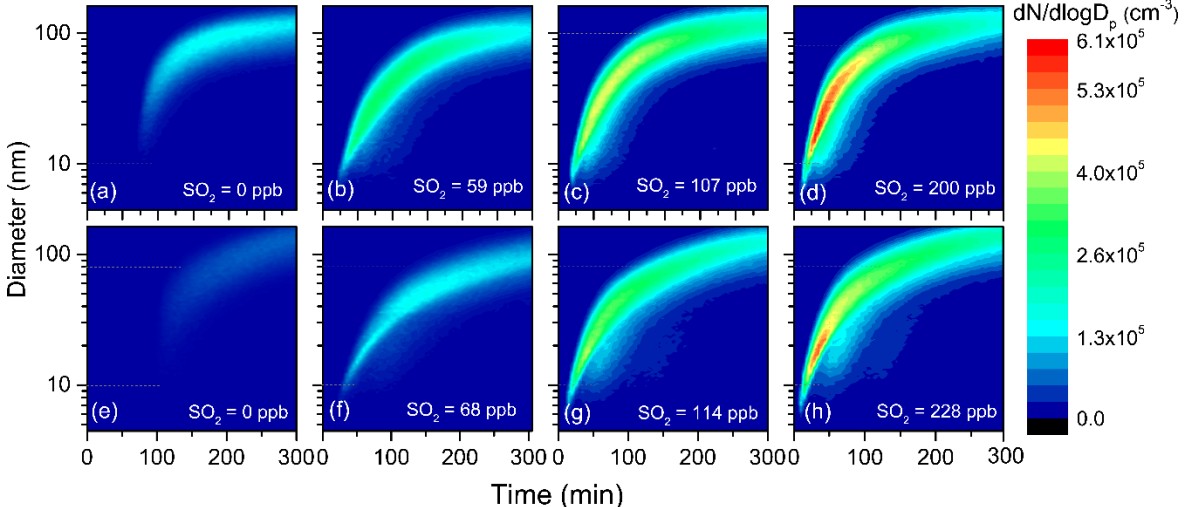

**Figure 1.** Evolutions of the number distributions of aerosol particles generated from TMB photooxidation in low-NO$_x$ (Panels a–d) and high-NO$_x$ (Panels e–h) experiments.

To fully give account for effects of changes in SO$_2$ emissions on aerosol formation, the key particle phase parameters (i.e., nucleation time, initial growth rate, and particle mean diameter) as a function of SO$_2$ levels are further compared in Fig. 2. In the present study, the SMPS instrument can measure particle larger than 4.5 nm and, therefore, the nucleation time here is defined as the time required for new secondary aerosols to grow to 4.5 nm after the lights have been turned on (Wyche et al.,

2009). The particle size rapidly increased within 30 min after nucleation, and gradually reached a stable level within 300 min photooxidation (Fig. 1). Consequently, the initial growth rate (GR$_{inital}$), calculated based on the method of Kulmala et al. (2012), is defined as the particle growth rate within 30 min after nucleation (Li et al., 2018). The mean diameter reported in this work represents the particle mean diameter measured at 300 min in each experiment. From Fig. 2, a significant negative

correlation was found between nucleation time and initial SO$_2$ level. Furthermore, when similar

amounts of $SO_2$ were introduced to the reaction mixture, the gap between the nucleation time of low-$NO_x$ and high-$NO_x$ would be reduced, which is in agreement with a previous study (Zhao et al., 2018). Under $SO_2$-free condition, new secondary aerosol could be generated by homogeneous nucleation involving key intermediate products of TMB oxidation. The delay time for particle

formation largely corresponds to the time required for intermediate products to build to sufficient concentrations in such a way that their saturation vapor pressure relative to the particle phase is exceeded. New secondary aerosol consists of later stage oxidation products, which might be also responsible for the delayed occurrence. It was demonstrated that higher OH concentration in the chamber could result in faster particle formation (Sarrafzadeh et al., 2016). However, the average

OH concentration in $SO_2$-free experiments is comparable with that in $SO_2$-added experiments (Table 1). There has been a gradual fall in the mixing ratio of $SO_2$ (Fig. S3) that could be oxidized to sulfuric acid ($H_2SO_4$) during TMB photooxidation. The formed $H_2SO_4$ could induce nucleation and increase the nucleation rate (Zhao et al., 2018; Blair et al., 2017), and these processes are responsible for the short nucleation time observed in the TMB/$NO_x$/$SO_2$ regime (Wyche et al., 2009). The mean

diameter of secondary aerosol decreased by 4–10 nm when 60–70 ppb of $SO_2$ was included in the matrix (Fig. 2). In contrast, at high-$SO_2$ levels ([$SO_2$]$_0$ > 100 ppb), increase in the initial $SO_2$ concentration led to an increase in particle mean diameter, regardless of low- or high-$NO_x$. The initial growth rate also showed a similar dependence on the $SO_2$ level as presented in Fig. 2. The nonlinear response of the particle mean diameter to $SO_2$ initial level is similar to the findings of

Julin et al. (2018), who found the response of particle size distribution to $NH_3$ emissions to be also nonlinear. Aerosol particles can grow in different ways such as gas-particle partitioning of semi-volatility organic compounds (SVOCs). Since the evaporation of SVOCs is important after partitioning to the particle phase, the rate at which SVOCs participate in the particle growth is lower than their condensation rate. However, recent advances give an insight that the particle-phase

chemistry such as heterogeneous reactions of SVOCs are substantially pronounced for the particle growth (Shiraiwa et al., 2013; Paasonen et al., 2018; Apsokardu and Johnston, 2018). Organosulfates can be produced by particle-phase reactions involving interactions between organics and inorganics. In this work, organosulfates were only detected in $SO_2$-involved photooxidation, indicating that additional particle-phase reactions can occur under $SO_2$-involved conditions.

Increasing the initial $SO_2$ level could induce the formation of more sulfate (Fig. S4) and the enhancement in the particle acidity during photooxidation (Liu et al., 2016; Kroll et al., 2006). The elevated particle acidity can promote more SVOCs to transform into low-volatile products such as organosulfates in the particle phase, thereby promoting the particle growth (Lin et al., 2014; Lal et al., 2012). Then, additional SVOCs could be further transferred from the gas phase to the particle

phase to increase the particle size. However, the particle mean diameter in low-$SO_2$ ([$SO_2$]$_0$ < 100 ppb) experiments is smaller than that in $SO_2$-free experiments. Our result is in line with the study of Wyche et al. (2009), who attributed this phenomenon to the larger number of particles produced under $SO_2$-involved condition. The presence of 59 ppb $SO_2$ caused the maximum number

concentration of particles to increase by $8.5 \times 10^4$ cm$^{-3}$ under low-NO$_x$ condition. When the SO$_2$ level increased from 0 to 68 ppb in high-NO$_x$ experiments, the corresponding particle number concentration increased from $2.9 \times 10^4$ to $9.3 \times 10^4$ cm$^{-3}$. Therefore, the amounts of products that condensed onto each aerosol particles significantly decreased in low-SO$_2$ experiments, which could result in the decrease in particle diameter (Liu et al., 2015a). The promoting effect of particle-phase chemistry on the particle size growth may not offset the inhibiting effect of the emergence of large number of particles on the particle size growth, thereby leading to the low particle diameter in low-SO$_2$ experiments.

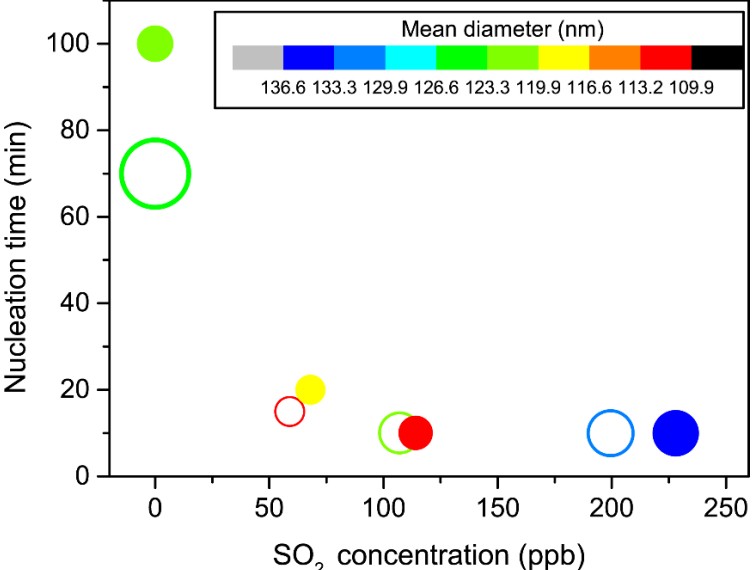

**Figure 2.** Particle nucleation time as a function of initial SO$_2$ concentration under low-NO$_x$ (open circles) and high-NO$_x$ (solid circles) conditions (Exps. 1–8). The symbol color indicates the particle mean diameter and symbol size represents the particle initial growth rate. Values of the particle parameters are listed in Table S1.

### 3.1.2 SOA yield in SO$_2$-added photooxidation

The particle volume concentration as a function of TMB consumption is presented in Fig. 3, where it can be observed that the particle formation increased with increasing initial SO$_2$ levels regardless of low- or high-NO$_x$ conditions. The present result is consistent with a previous research, which found that limonene SOA formation is significantly promoted with SO$_2$ addition (Ye et al., 2018). SO$_2$ was found to perturb particle formation by inducing chemical reactions in the gas- and particle-phase (Wang et al., 2019a; Friedman et al., 2016). We are unable to fully rule out the SO$_2$ impacts on gas-phase chemistry. However, the decay of TMB was essentially unchanged when SO$_2$ was introduced into the chamber (Fig. S5), which suggests the unlikeliness of SO$_2$ addition to affect the gas-phase chemistry of TMB photooxidation (Kleindienst et al., 2006). Instead, it is more likely

attributed to the formation and condensation of $H_2SO_4$ and/or the enhancement of organic aerosols formation. We assumed full conversion of the consumed $SO_2$ into $H_2SO_4$ aerosol particles and found that the contribution of the formed $H_2SO_4$ to the increase in particle volume concentration was less than 100% (See Sect. S2). In addition, pure $SO_2$ oxidation experiments without TMB addition also indicated that the enhancement in aerosol particles by $SO_2$ introduction cannot be solely attributed to inorganic aerosol formation (See Sect. S2). To calculate the net SOA yield, the inorganic mass concentration was subtracted from the particle mass concentration based on IC measurements of generated particles. The influence of $SO_2$ initial level on SOA yield can be seen in Table 1 as high $SO_2$ levels contribute to produce somewhat high SOA yields. Figure 4 compares the SOA yields obtained from the present work with those found in previous studies with similar experimental conditions. The SOA yields from the two $SO_2$-free experiments are comparable to that reported from the study of Liu et al. (2012) and fit quite well with the yield curve of Odum et al. (1996). The SOA yields in our TMB/$NO_x$ photooxidation experiments were 3.8% and 3.5%, which were closed to 3.9% and 4.2% derived from yield curve of Odum et al. (1996) under same mass concentration. In contrast, with similar mass concentration, the SOA yields in $SO_2$-added regimes were higher than those in previous studies (Odum et al., 1996; Liu et al., 2012). Here the neutralization degree of particle, which was calculated as the molar ratio of $NH_4^+$ to the sum of $SO_4^{2-}$ and $NO_3^-$ (Lin et al., 2013), was used as a tool to roughly estimate the aerosol acidity of collected particle samples. The value of neutralization degree were lower than 1 in this work, indicating acidic aerosols (Lin et al., 2013). The increase in aerosol acidity could be largely responsible for the observed enhancements in SOA formation in $SO_2$-invloved experiments. The OH oxidation of TMB can result in the formation of multifunctional carbonyl compounds (Liu et al., 2012; Zaytsev et al., 2019), which could promote SOA formation via acid-catalyzed heterogeneous reactions. In addition, the particle surface area concentrations significantly increased with increasing $SO_2$ initial concentrations in both low-$NO_x$ and high-$NO_x$ conditions (Table 1), which might also result in the enhancement in the SOA yield. Besides gas-particle partitioning of SVOCs, the fates of SVOCs in the chamber also include chemical reactions and chamber wall losses. Therefore, in the batch-mode chamber experiments, the gas-particle partitioning of SVOCs have a great sensitivity to particle surface areas (Zhang et al., 2015; Han et al., 2019). Recently, Zhao et al. (2018) examined the $SO_2$ effects on the SOA formation and suggested that providing additional particle surfaces by $SO_2$-induced new particle formation leads to the increase in SOA yield. The effects of the particle surface area concentration on organic aerosol formation were explored by Han et al. (2019), who also found that increasing the particle surface area concentrations can significantly increase the organic aerosol mass yield due to greater partitioning of semi-volatility organic products to the particle-phase. Increasing the particle surface area can limit the gas-wall interactions of organic vapors and is favorable for the movement of more SVOCs from the gas phase to the particle side (Han et al., 2019). These additional SVOCs can also undergo further particle chemistry such as acid-catalyzed heterogenous reactions to strongly enhance aerosol particle formation in TMB/$NO_x$/$SO_2$ photooxidation (Apsokardu and

Johnston, 2018).

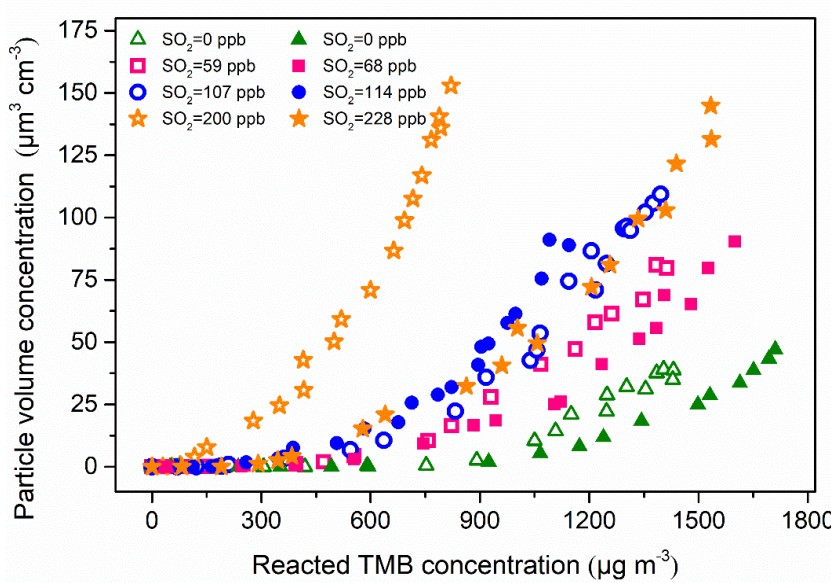

**Figure 3.** Growth of particle volume concentrations from TMB photooxidation as a function of
TMB consumption for eight experiments with different initial $SO_2$ concentrations (Exps. 1–8). The
open symbols and solid symbols represent low- and high-$NO_x$ experiments, respectively.

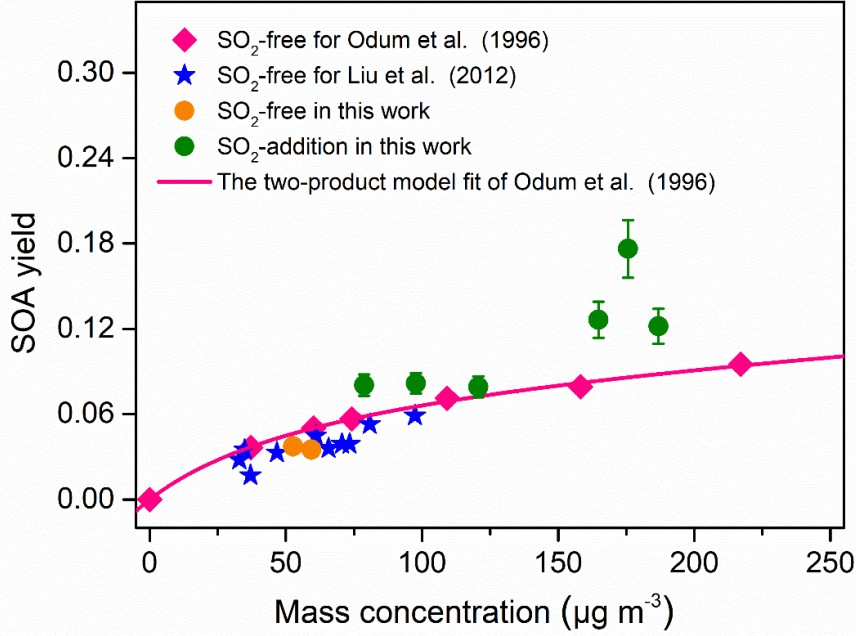

**Figure 4.** Comparison of SOA yields from TMB photooxidation as a function of SOA mass
concentration with data reported from literature. The SOA density in the whole study was assumed
to be 1.4 g $cm^{-3}$. The error bars represent errors in the SOA yield results and the errors were
calculated from error propagation using the sum of the uncertainties in TMB data and the systematic
error of SMPS. The SOA yields for Liu et al. (2012) were extracted from Table 1 in the study. The
pink line is the best fit two-product model for the SOA yields and was extracted from Figure 2 in
the work of Odum et al. (1996).

### 3.1.3 Particle chemical composition in SO$_2$-added photooxidation

In order to investigate the effects of SO$_2$ on the chemical composition of aerosol particles, the particles were first characterized by ATR-FTIR. Figure 5 compares the characteristic ATR-FTIR spectra of particles formed from the photooxidation of TMB under different conditions and the detailed information on the assignment of absorption peaks are given in Table S2. For the samples collected from TMB/NO$_x$ photooxidation, the particles exhibited an O-H stretch at 3600–3000 cm$^{-1}$ as shown in Fig. 5(a) and (b). The C=O stretch of carbonyl at 1720 cm$^{-1}$ suggests that aldehydes, ketones, and carboxylic acids are significant particle components, while the absorptions at 844 cm$^{-1}$ (NO symmetric stretch), 1284 cm$^{-1}$ (NO$_2$ symmetric stretch), and 1647 cm$^{-1}$ (NO$_2$ asymmetric stretch) are the characteristic peaks of organic nitrates. The slight absorbance at 941 cm$^{-1}$ indicates the presence of peroxides containing O-O groups. Interestingly, compared with the particles from TMB/NO$_x$ experiments, there was a new peak at 615 cm$^{-1}$ for the particles generated from TMB/NO$_x$/SO$_2$ experiments, regardless of low- or high-NO$_x$ conditions. The 615 cm$^{-1}$ peak is characteristic of inorganic sulfates, and further highlights that the presence of SO$_2$ promotes the formation of inorganic sulfates as pointed out in previous studies (Chen et al., 2019; Liu et al., 2016). The strong absorption at 1081 cm$^{-1}$ arises from the S=O bands in the particle components. Field and laboratory studies have reported that inorganic sulfate could convert into organosulfur compounds in the atmosphere (Riva et al., 2019; Nestorowicz et al., 2018). Therefore, the peak at 1081 cm$^{-1}$ may be mainly from the absorption of both inorganic sulfates and organosulfur compounds.

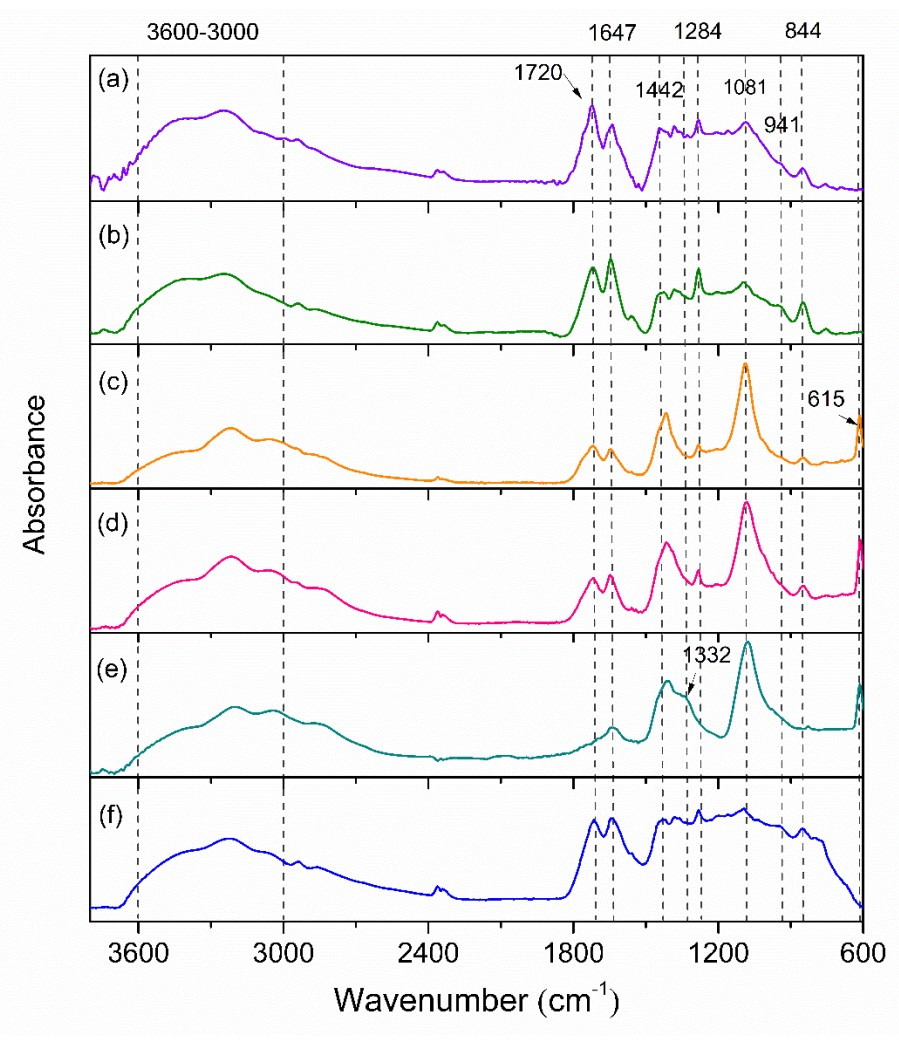

**Figure 5.** ATR-FTIR spectra of aerosol particles generated from TMB/NO$_x$ (a, Exp. 1; b, Exp. 5), TMB/NO$_x$/SO$_2$ (c, Exp. 4; d, Exp. 8), TMB/NO$_x$/NH$_3$/SO$_2$ (e, Exp. 12), and TMB/NO$_x$/NH$_3$ (f, Exp. 10) photooxidation.

The chemical compositions of particles generated in TMB/NO$_x$ and TMB/NO$_x$/SO$_2$ photooxidations were further measured with high-resolution mass spectrometry (HRMS), to determine whether organosulfur compounds were formed in SO$_2$-added experiments. The formation of organosulfur compounds was recognized by the oxidation mechanism and the loss of characteristic fragment ions at $m/z$ 79.95 (SO$_3^-$), 80.96 (HSO$_3^-$), and 96.96 (HSO$_4^-$) in MS/MS spectra (Figs. S7-S8). The list of compounds observed in this work along with molecular weights (MW), measured masses and proposed structures are presented in Table S3. In SO$_2$-free experiments, the major aerosol components were multifunctional alcohols, peroxides, organic nitrates, ketoaldehydes, and ketocarboxylic acids. With the same analytical methods, the same products were also observed in SO$_2$-added photooxidation. The most striking result to emerge from Table S3 is that ten organosulfates (OS-214, OS-226, OS-228, OS-240, OS-242, OS-244, OS-268, OS-300, OS-316, and OS-345) and two organic sulfonates were only detected in filter samples collected from the

TMB/NO$_x$/SO$_2$ photooxidation experiment, indicating that SO$_2$ emissions in the atmosphere can alter the aerosol formation chemistry and thus influence the aerosol chemical composition. To the best of our knowledge, this is the first time the ten organosulfates are identified in TMB/NO$_x$/SO$_2$ photooxidation experiments. Recently, some sulfur-containing compounds from field measurements were designated as compounds of unknown origin. For example, the MW 214 organosulfate has been detected in PM$_{2.5}$ collected from the highly polluted megacity Shanghai (Cai et al., 2020). However, the VOC precursor for this organosulfate formation was not reported. The MW 242 organosulfate found in Baengnyeong Island was also classified as organosulfur of unknown origin (Boris et al., 2016). O'Brien et al. (2014) found the formation of MW 228 organosulfate in ambient aerosol particles but its specific source was not pointed out. Evidence from this study suggests that TMB photooxidation in the presence of SO$_2$ might contribute to the formation of these organosulfates (MW = 214, 228, and 242) in the ambient air. Importantly, an organosulfur compound with a formula of C$_7$H$_{12}$O$_7$S (MW = 240) was observed in ambient fine aerosols and was tentatively assigned to an oxidation product of anthropogenic 1,3,5-trimethylbenzene (Boris et al., 2016). The finding of the current study suggests that the organosulfur compound (MW = 240) may also be produced from the photooxidation of 1,2,4-trimethylbenzene in the presence of SO$_2$. In addition, the MW 226, 240, and 268 organosulfur compounds were designated as biogenic-derived organosulfates in previous field studies (Cai et al., 2020; Boris et al., 2016). More recently, Chen et al. (2020b) suggested that heterogeneous OH oxidation of isoprene-derived SOA can contribute to the formation of an organosulfate with molecular weight at 228. Our results show the detection of OS-226, OS-228, OS-240, and OS-268 organosulfates, which are isomers of organosulfates derived from isoprene (Cai et al., 2020), isoprene (Chen et al., 2020b), limonene (Cai et al., 2020), and limonene (Boris et al., 2016), respectively. More studies need to be undertaken to differentiate various sources of organosulfates in the ambient aerosols through chemical synthesis of authentic organosulfates standards.

The mechanisms describing the formation of OS-226, OS-228, OS-240, OS-242, OS-244, OS-300, OS-316, and OS-345 are proposed in Fig. 6. Following analogous mechanisms for toluene photooxidation, the oxidation of TMB is dominantly initiated by OH addition to the benzene ring to form TMB-OH adduct, which can react with O$_2$ through recombination to produce bicyclic peroxy radical. It has been established that a series of ring-retaining (product A in Fig. 6) and ring-opening products (products B, C, and D in Fig. 6) can be generated by the further reaction of bicyclic peroxy radical in the presence of NO (Zaytsev et al., 2019; Li and Wang, 2014). Both ring-opening and ring-retaining compounds are expected to contribute significantly to organosulfate production. Here, we take unsaturated ketoaldehyde (product C in Fig. 6) as an example to describe the possible formation mechanism of organosulfate observed in the present study. The reaction of OH with compound C involves OH addition to unsaturated C=C bonds to form an alkyl radical, which can react subsequently with O$_2$ to yield organic peroxy radical. Further reactions of organic peroxy

radical can follow two different pathways. One pathway is that the organic peroxy radical undergoes a 1,5-H-shift isomerization to form a new acyl radical. Insight from a previous review suggested that acyl radical react with $O_2$ to yield acylperoxy radical, which could further react with $HO_2$ to form multifunctional hydroperoxide (Ziemann and Atkinson, 2012). The second channel is the reaction of organic peroxy radical with $HO_2$ to produce hydroperoxide, terminating directly the radical chain. Acid-driven heterogeneous chemistry of hydroperoxide has been previously adopted to explain the generation of certain OSs (Riva et al., 2016a; Riva et al., 2016b). With the presence of sulfuric acid formed by the oxidation of $SO_2$ by OH, TMB-derived hydroperoxides can be hydrolyzed by $H^+$ and then react with inorganic $SO_4^{2-}$ to form organosulfates. The conversion of inorganic sulfates to organosulfates could cause changes in aerosol growth, multiphase chemistry, and acidity (Zhang et al., 2019; Riva et al., 2019).

**Figure 6.** Proposed mechanisms for organosulfate formation from the photooxidation of TMB in the presence of $SO_2$. The black boxes mark the ring-opening and ring-retaining products suggested in previous studies (Li and Wang, 2014; Zaytsev et al., 2019). The compounds in red are organosulfates detected by UPLC-HRMS in this work.

The MW 228 and 230 organic sulfonates were assigned as sulfonates containing an aromatic ring based on accurate mass measurements and comparison of mass fragmentation patterns with other aromatic sulfonates (Riva et al., 2015a). MS/MS spectra and the proposed fragmentation schemes of sulfonates are reported in Fig. S8. For MW 228 sulfonate, the MS/MS spectra showed the fragment ions at $m/z$ 79.95721 ($SO_3^{\cdot-}$), 118.96625 ($C_8H_7O^{\cdot}$, M - $SO_3^{\cdot-}$ - CO), and 163.04025 ($C_9H_7O_3^-$, M - $SO_2$) as presented in Fig. S8. For the MW 230 sulfonate, the fragment of parent ion

at $m/z$ 229.01706 could occur by the loss of 44 mass units to give the product ion at $m/z$ 185.02777
(Fig. S8). The loss of 79.95747 mass units as sulfite radical is in accord with the MS/MS spectra of
aromatic sulfonates generated from the photooxidation of polycyclic aromatic hydrocarbons (Riva
et al., 2015b). A recent field measurement demonstrated that aromatic organosulfur compounds
account for a substantial fraction of total organosulfur compounds in Shanghai, China, highlighting
the importance of aromatic organosulfur compounds (Ma et al., 2014). Aromatic sulfonates
formation from the photooxidation of TMB in the presence of $SO_2$ was unexpected and the exact
formation pathway of the aromatic sulfonates warrants further investigation.

### 3.2 Effects of $NH_3$

### 3.2.1 Particle formation and growth in $NH_3$-involved photooxidation

High-$NO_x$ photooxidation experiments were carried out in the presence/absence of $NH_3$. The
average OH concentrations were similar for each experiment within 300 min of irradiation (Table
1). Figure 7 displays the volume and number concentrations of aerosols as a function of time in
different photooxidations (Exps. 5, 8–12). TMB was oxidized to produce many secondary aerosols
under continuous UV irradiation. The volume concentrations of aerosol particles have a clear
positive correlation with $NH_3$ initial level for all conditions. However, the effect of $NH_3$ on particle
formation was not as pronounced as that of $SO_2$ with similar concentration (Fig. 7). In
TMB/$NO_x$/$NH_3$ photooxidation, the net SOA yield increased slightly from 3.5% to 5.1% as $NH_3$
initial level increased from 0 to 200 ppb (Table 1). Our result is consistent with the finding of Chen
et al. (2020a), who showed that $NH_3$ did not significantly affect SOA formation from toluene/$NO_x$
photooxidation under dry condition. Interestingly, SMPS measurements demonstrated that the
coexistence of $SO_2$ and $NH_3$ can considerably promote secondary aerosol formation (Fig. 7). After
subtracting the inorganic components, it was seen that the net SOA yield could increase to 13.7%
with the introduction of 200 ppb $NH_3$ and 234 ppb $SO_2$, indicating the synergetic effects of $NH_3$ and
$SO_2$ (Chu et al., 2016). The flux of the gas-phase products diffusing to a particle partly depends on
the surface area of the particle. The coexistence of $SO_2$ and $NH_3$ promoted the increase in particle
surface area concentrations (Table 1). The ability of particle formation originating from gas-to-
particle conversion may be significantly stronger with $SO_2$ and $NH_3$ introduction, leading to the
enhancement in particle formation. The total number concentrations of aerosol particles increased
rapidly in all experiments after nucleation was initiated as shown in Fig. 7 (b). After reaching the
maximum concentration, the number density of particles declined gradually because of particles
coagulation and deposition to the chamber walls. Increasing the $NH_3$ level to 200 ppb enhanced the
particle maximum number concentration by factors of 2.0 and 1.7 under $SO_2$-free and $SO_2$-involved
conditions, respectively. Our results are valuable in terms of the potential of $NH_3$ emission
reductions to improve air quality by decreasing total particle number concentrations.

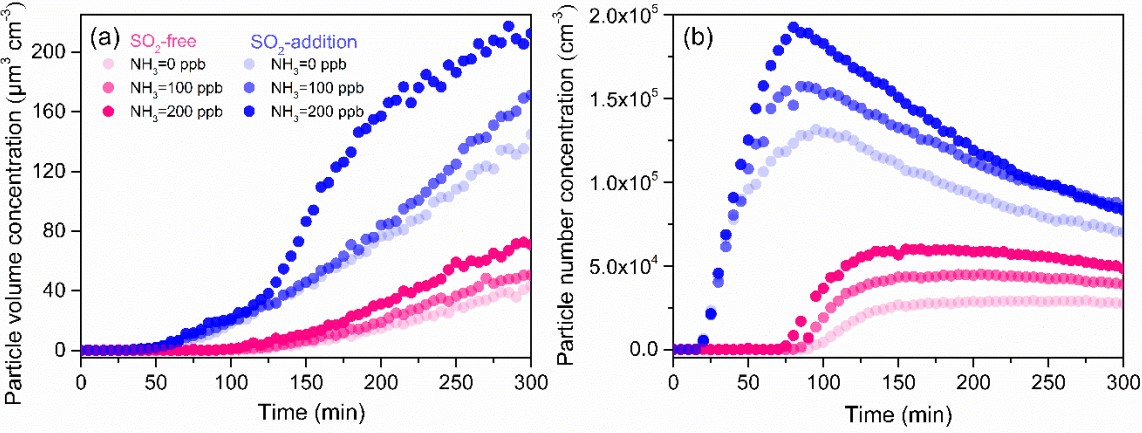

**Figure 7.** Time evolutions of the volume (a) and number (b) concentrations of aerosol particles from TMB photooxidation with different initial NH$_3$ levels under SO$_2$-free and SO$_2$-addtion (~ 230 ppb) conditions.


Once generated, aerosol particles need to grow to a larger size (> 50–100 nm) before they exert significant impact on global climate and public health. To explore the effects of NH$_3$ on particle growth, the initial growth rate of aerosol particles from different experiments are also compared in Fig. S9. In SO$_2$-free experiments, the increase in NH$_3$ initial concentrations led to remarkable

increase of the initial growth rate of aerosol particles. High initial growth rates were also found in the photooxidation of other aromatics such as toluene and o/m/p-xylene with NH$_3$ addition (Li et al., 2018). This result may be explained by the fact that under SO$_2$-free condition, NH$_3$ mainly reacts with acids to produce ammonium salts. Previous studies have reported that ammonium salts could partition into the initial growth process of new secondary aerosol particles and thus increase the

particle initial growth rate (Li et al., 2018; Zhu et al., 2014). More interestingly, NH$_3$ level did not substantially affect the initial growth rate of particles in the presence of SO$_2$ (Fig. S9). The results in Fig. 7 have demonstrated that the synergetic effects of NH$_3$ and SO$_2$ can promote new particle formation (Lehtipalo et al., 2018). In SO$_2$-involved experiments, NH$_3$ molecules tend to promote new particles formation rather than particles growth. Note that these effects of NH$_3$ on particle initial

growth might be more complex in the ambient air with high levels of SO$_2$, NH$_3$, NO$_x$, VOCs than current smog chamber experiments.

### 3.2.2 Particle chemical composition in NH$_3$-involved photooxidation

The ATR-FTIR spectra of aerosol particles from NH$_3$-added experiment are also given in Fig. 5. A

previous study suggested that the absorbances at 3310–3360 cm$^{-1}$ and 1550–1650 cm$^{-1}$ can be assigned to N-H stretch and C-N-H bend in secondary amine molecules, respectively (Babar et al., 2017). However, no evidence of secondary amine formation was detected and no clear FTIR spectra

differences between aerosols from TMB/NO$_x$ experiment and aerosols from TMB/NO$_x$/NH$_3$ photooxidation were found in this work (Fig. 5). It has been recently discovered that the conversion of oxidized organics to nitrogen-containing compounds in the presence of NH$_3$ is more likely to occur in high RH condition (Zhang et al., 2020). NH$_3$ uptake by TMB-derived aerosol particles may be limited to the aerosol surface under low RH condition (RH < 20%) (Bell et al., 2017). The amounts of secondary amine compounds formed from the NH$_3$ uptake by aerosol may be small, resulting in no characteristic peaks of secondary amine in ATR-FTIR spectra. Surprisingly, the effects of NH$_3$ on aerosol chemical compositions were increasingly important under SO$_2$-rich condition. As displayed in Fig. 5(e), the strong C=O band at 1720 cm$^{-1}$ converted to a shoulder upon NH$_3$ addition to the TMB/NO$_x$/SO$_2$ reaction system. In addition, the C-N stretch at 1332 cm$^{-1}$ was observed in TMB/NO$_x$/SO$_2$/NH$_3$ photooxidation (Fig. 5). The particle-phase C-N stretch has been also characterized in m-xylene photooxidation experiments with NH$_3$ addition and was suggested to account for nitrogen-containing compounds (Liu et al., 2015b). A smog chamber study also found that the coexistence of SO$_2$ and NH$_3$ substantially enhanced the formation of nitrogen-containing compounds from the photooxidation of toluene (Chu et al., 2016). In the current work, the presence of SO$_2$ could promote the increase in particle acidity. Elevating particle acidity could facilitate the reaction of NH$_3$/NH$_4^+$ with carbonyl-containing compounds, leading to the formation of nitrogen-containing organic compounds (Liu et al., 2015b).

**Figure 8.** Simplified formation mechanism of the detected products in SOA formed from NH$_3$-involved photooxidation. Some observed products are produced through same chemical mechanism and for simplicity, only one product is drawn in this figure as an example.

The MS measurement results are consistent with FTIR analysis. As shown in the MS spectra of aerosol samples (Fig. S10), under SO$_2$-free condition, the presence of NH$_3$ did not result in considerable changes in peak numbers and abundance for both positive ion mode and negative ion mode. NH$_3$ could slightly enhance SOA formation in SO$_2$-free experiment as mentioned in Sect. 3.2.1. Therefore, the NH$_3$-induced changes in the absolute concentrations of organic components

might be small in $SO_2$-free experiments, leading to similar mass spectra for Fig. S10(a) and Fig. S10(b). In addition, the major products (Table S4) are likely generated by similar chemical mechanisms (Fig. 8), which are not sensitive to the change in initial $NH_3$ levels under current

experimental conditions. First, the photooxidation of TMB in the presence of NO can result in the formation of bicyclic oxy radicals that could decompose to a series of ring-opening products including biacetyl, epoxy-dicarbonyl, and carbonylic products (Li and Wang, 2014). As depicted in Fig. 8, the first-generation products could be further oxidized by OH. For example, oxidation of methylglyoxal by OH can proceed through abstraction of the aldehydic hydrogen to form the

$CH_3C(O)C(O)\cdot$ radical, which may react with $O_2$ and then with $HO_2$ to form pyruvic acid. Second, bicyclic peroxy radical (BPR) is a key intermediate for the formation of non-aromatic ring-retaining products. Reactions of BPR with $R'O_2$ can form either bicyclic carbonyl and bicyclic alcohol, which further undergoes OH oxidation to yield the $C_9H_{16}O_6$ and $C_9H_{16}O_9$ compounds. BPR can produce bicyclic organonitrates by reaction with NO, and can also undergo intramolecular H-shift followed

by $O_2$ addition to form a new bicyclic peroxy radical. The new bicyclic peroxy radical reacts with $HO_2$ to generate highly oxygenated organic molecules, consistent with a recent study (Wang et al., 2020b). To our knowledge, $NH_3$ does not basically affect the reaction of free radicals in gas-phase during the photooxidation of TMB. Generally, $NH_3$ levels play a negligible role in the aerosol organic composition in TMB photooxidation without $SO_2$ addition. In contrast, under $SO_2$-rich

condition, the increase in $NH_3$ level led to a significant increase in the abundance of organic compounds (especially for compounds with $m/z > 200$) in both positive and negative ion modes (Fig. S11). The introduction of $SO_2$ and $NH_3$ lead to the formation of ammonium sulfate (Fig. S12), which is an attractive condensation sink for organic vapors. High particle surface area concentration in $TMB/NO_x/SO_2/NH_3$ experiments may increase the abundance of organic compounds in the bulk

phase. To better explain this effect, the saturation mass concentrations of detected products were predicted based on a previous method (Li et al., 2016) and the calculated results are shown in Fig. 9 and Table S4. During the photooxidation of TMB, the fates of organic compounds are mainly governed by the competition between fragmentation and functionalization. Losing carbon atoms increases product volatility, which could be partly compensated by functionalization. Among the

compounds present in aerosol particles formed from $NH_3$-added systems, fourteen products were C9 or smaller multifunctional oxidation products. From Fig.9, the range of product saturation mass concentration spanned approximately 8 orders of magnitude, indicating that the measured particle-phase products are considerably different regarding volatility. The products in the particle-phase are classified into three classes in Fig. 9: low-volatility organic compound, intermediate volatility

organic compounds, and semi-volatility organic compounds. The measured products might not be responsible for homogenous nucleation but these compounds can gradually condense onto nucleation particles (i.e., ammonium sulfate) to contribute to aerosol formation and growth, which highlights the role of ammonium sulfate in this case.


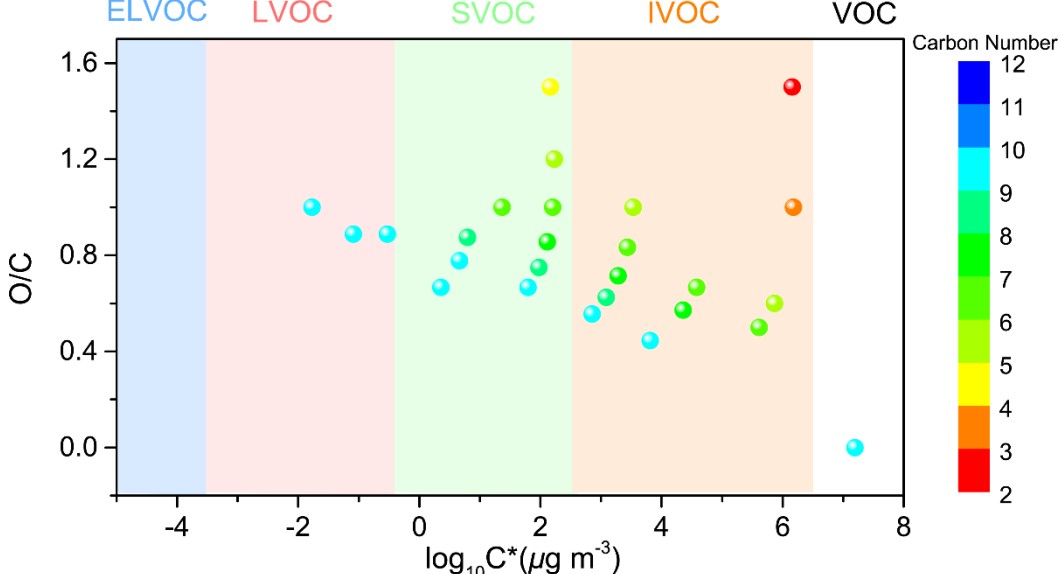

**Figure 9.** TMB and detected products (Table S4) displayed in the two-dimensional volatility-oxidation space. Based on the method of Li et al. (2016), we grouped the detected compounds in the classes of extremely low-volatility organic compound (ELVOC), low-volatility organic

compound (LVOC), semi-volatility organic compound (SVOC), intermediate volatility organic compounds (IVOC), and volatility organic compound (VOC).

**4 Conclusions**

In summary, we explored the detailed effects of $SO_2$ and $NH_3$ on secondary aerosol formation from

TMB photooxidation. Our results demonstrate a substantial increase in ultrafine particle (< 100 nm) number concentrations resulting from $SO_2$ addition. Significant increases in SOA yields were found in TMB/$NO_x$/$SO_2$ photooxidation, due to acid-driven heterogeneous reaction. The laboratory characterization of SOA composition confirmed the formation of new organosulfates at MW 214, 226, 228, 240, 242, 244, 268, 300, 316, and 345. The MS data give experimental evidence that the

MW 214 and 242 organosulfates could account for organosulfates previously designated as unknown origin in ambient PM$_{2.5}$, while some of them that were observed in TMB/$NO_x$/$SO_2$ photooxidation are isomers of recognized biogenic organosulfates. This indicates that care must be taken in the identification of TMB-derived organosulfates in ambient aerosols. More laboratory and field studies with organosulfate authentic standard should be carried out to determine the accurate

yield of the measured organosulfates and the contribution of TMB-derived organosulfate to total atmospheric organosulfate. In addition, the composition of secondary aerosol could determine the physicochemical properties of aerosol particles (e.g. viscosity and phase state). Changes in $SO_2$ emissions in different regions all over the world have great implications for the physicochemical properties of aromatics-derived SOA and, thus, highly influence the global climate.

The presence of $NH_3$ also increased the number and volume concentration of secondary aerosol particles, especially under $SO_2$-rich condition. We characterized a series of multifunctional ring-retaining and ring-opening organic compounds containing one and even more carbonyl and alcohol. The predicted volatility distributions of products suggested that the measured later-generation products progressively condense onto nucleation particles to enhance particle formation in $NH_3$-

added photooxidation. Current models that are used to assess aerosol-climate interactions should fully take into account the influence of $NH_3$ on secondary aerosol formation, which is especially significant in regions with strong $NH_3$ emissions.


*Data availability.* Experimental data are available upon request to the corresponding author.

*Supplement.* The supplement related to this article is available online at:

*Author contribution.* Zhaomin Yang: Conceptualization, Methodology, Investigation, Validation, Visualization, Writing – original draft preparation. Li Xu: Investigation. Narcisse T. Tsona: Visualization, Writing – review & editing. Jianlong Li: Visualization, Writing – review & editing. Xin Luo: Methodology, Resources. Lin Du: Conceptualization, Investigation, Funding acquisition, Project administration, Resources, Supervision, Visualization, Writing – review & editing.

*Competing interests.* The authors declare that they have no conflict of interest.

*Financial support.* This work was supported by National Natural Science Foundation of China (91644214), Youth Innovation Program of Universities in Shandong Province (2019KJD007), and Fundamental Research Fund of Shandong University (2020QNQT012).

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
