# Peer review of "SO2 and NH3 emissions enhance organosulfur compounds and fine particles formation from the photooxidation of a typical aromatic hydrocarbon"

_Atmospheric Chemistry and Physics, 2021_

## Author Comment (AC1)

We sincerely thank the Referee for the valuable comments. Our manuscript has been revised according to the comments from the Referee and our responses to the comments are as follows. For clarity, the comments are reproduced in blue, authors' responses are in black and changes in the manuscript are in red.

**Responses to Anonymous Referee #1**

This smog chamber study investigated the effects of $NH_3$, $SO_2$, and $NO_x$ on SOA formation under UV irradiation from 1,2,4-trimethylbenzene (TMB) over a period of 5-6 hours. The smog chamber was monitored in real time using a scanning mobility particle sizer; gas analyzers for $SO_2$, $NO_x$, $NH_3$, and $O_3$; and gas chromatography-flame ionization for TMB. Offline analysis used infrared spectrophotometry, ion chromatography, and liquid chromatography-mass spectrometry. The study found that the presence of $NH_3$ and $SO_2$ both individually and synergetically increase SOA yield, and that $SO_2$ speeds nucleation, possibly through uptake onto $H_2SO_4$ surfaces. New organosulfates were identified and reaction schemes and structures were proposed for some; some organosulfates had molecular weights consistent with TMB-derived aerosol components found in the atmosphere. $NH_3$ was found to react to form organic nitrogen compounds in the aerosol phase, but only in the presence of $SO_2$, attributed to formation of ammonium sulfate. Aerosol components had a wide range of volatility almost nine orders of magnitude, as predicted by elemental composition from mass spectrometry.

This study is a good match for the scope and aims of ACP, and it does an excellent job conveying its own novelty even if the reader is not well versed in aromatic aerosol chemistry. It represents a necessary contribution to the aerosol research community's understanding of $SO_2$ and $NH_3$ dynamics, and has implications for reactions of a large number of aromatic VOCs as well as human health in the findings regarding enhanced UFP fraction.

Major Comments

This article is solid, if not concise, and all sections contribute to the main point as a

cohesive whole. The introduction, methods section, and discussion of limitations are excellent, and while there is far more description of results than there is analysis of results I believe that both the analysis and discussion of atmospheric implications are sufficient. All of my suggestions for improvement are minor, relating to clarity and readability.

Minor Comments

line 13: Consider a brief explanation of the atmospheric relevance of TMB in the abstract.

**Author Reply:**

Aromatic hydrocarbons can dominate the VOC budget in certain urban areas with TMB being one of the most significant species (Ran et al., 2009). TMB is mostly emitted from anthropogenic sources such as solvent use (Mo et al., 2021). Recent studies have recognized TMB as an important species in SOA formation in the atmosphere (Wang et al., 2020; Zaytsev et al., 2019; Mehra et al., 2020). Hence, the following text has been added in the abstract to explain the relevance of TMB in the atmosphere.

Aromatic hydrocarbons can dominate the volatile organic compounds budget in the urban atmosphere. Among them, 1,2,4-trimethylbenzene (TMB), mainly emitted from solvent use, is one of the most important secondary organic aerosols (SOA) precursors.

line 69-70: "Equivocally not originated from biogenic VOCs" is a little unclear. Possibly change to "unidentified OSs with C2-C25 skeletons that may not have originated from biogenic VOCs" or similar.

**Author Reply:**

We have modified the sentence in the revised manuscript as: Recent field studies reported that some unidentified OSs with $C_2$–$C_{25}$ skeletons may not be originated from biogenic VOCs.

line 254-55: Make it clear whether ammonium sulfate particles were introduced during experiments or as an independent wall loss experiment.

**Author Reply:**

Ammonium sulfate (AS) seed particles were not introduced into the chamber over the course of particle formation experiments. We used inert AS particles only in independent wall loss experiments in order to characterize the wall loss rate of AS particles, which was adopted to correct aerosol particle concentrations. This particle wall-loss correction method has been commonly used in previous studies (Chen et al., 2019; Charan et al., 2020). For clarity, we have included the following sentences in the revised manuscript.

Page 5, lines 168–169

Seed particles were not introduced into the chamber over the course of particle formation experiments.

Page 10, line 275

In order to determine the particles wall loss rates, we carried out independent wall loss experiments.

Is this dependent on humidity or hygroscopicity of the particles? Not VERY important, but may be a limitation worth discussing.

**Author Reply:**

We agree with the Referee that particle wall depositions are affected by the physicochemical properties of particles to some extent. It should be noted that the description of an important step of the wall loss experiments was missing in our original manuscript. Indeed, the AS solution was added to a TSI atomizer (Model 3076) to produce droplets, which passed simultaneously through a silica gel diffusion dryer to produce dry AS particles. Although the hygroscopicity of AS particles is not totally similar to aerosol particles generated, the wall-losses of particles are commonly evaluated by independent seed experiments where inert AS particles are used (Chen et al., 2019; Charan et al., 2020).

Aerosol particles can deposit onto the chamber walls due to gravitational settling, Brownian diffusion, convection, and electrostatic effects (Crump and Seinfeld, 1981). The wall-loss rate of particles depends mainly on the smog chamber design parameters

(Charan et al., 2019). Previous studies showed that elevating RH inside the chamber did not remarkably influence the particle wall loss rates (Yu et al., 2011; Ge et al., 2017). Ge et al. (2017) used a same wall loss constant for the particle wall-loss corrections in experiments with RH ranged from 11% to 90%. Although the changes in RH can influence the hygroscopicity, density, size, and chemical composition of particles (Hinks et al., 2018; Yu et al., 2011), almost identical humidity ((25 ± 1) %) condition was achieved among each experiment in the current work. Therefore, we believe that the particle wall depositions are not basically influenced by humidity or hygroscopicity of the particles. The description about the wall loss experiments has been included in the revised manuscript as follow.

Page 10, lines 276-279

An aqueous solution of ammonium sulfate was fed to a constant output atomizer (Model 3706, TSI, USA) to produce droplets, which passed simultaneously through a silica gel diffusion dryer to introduce dry particles into the chamber. The size distributions of ammonium sulfate particles were measured by SMPS for 480 min. The wall losses of particles are size-dependent and, thus, we used a size-dependent particle wall-loss correction approach, which is described in detail in the supplement

line 284: Were these from two single experiments? If so, include experiment numbers. Otherwise I want to know the standard deviations for these values.

**Author Reply:**

Yes, the reported particle number concentrations were obtained in independent $SO_2$-free experiments with different initial $NO_x$ levels. We have highlighted the experiment numbers in the revised manuscript as:

After 300 min UV irradiation without $SO_2$ introduction, the total maximum number concentration of aerosol particles was only $2.7 \times 10^4 \, cm^{-3}$ and $2.9 \times 10^4 \, cm^{-3}$ under low-$NO_x$ (Exp. 1) and high-$NO_x$ (Exp. 5) experiments, respectively.

line 291-95: Before noting that UFP are more harmful to human health, you may want to make it explicit in the text analysis of figure 1 that increased $SO_2$ concentration

increases the fraction of UFP in the aerosol size distribution.

**Author Reply:**

The SMPS instrument could measure particles larger than 4.50 nm in this work. In order to more clearly analyze the effects of $SO_2$ on the ultrafine particle formation, the number concentrations of ultrafine particles in the range of 4.50–100 nm were summed up and they are shown in Fig. R1, where it can be seen that the total number concentration of ultrafine particles is significantly enhanced with increasing $SO_2$ levels. Therefore, the following text was added in the revised manuscript and Fig. R1 was added as Fig. S2 in the revised supplement.

Page 10, lines 309–315

Interestingly, the particle maximum number concentration considerably increased with increasing $SO_2$ levels, regardless of low- or high-$NO_x$ conditions (Table 1). As shown in Fig. 1, although the unimodal particle size distribution in the presence of $SO_2$ was similar to that in its absence, particles with mobility diameter ranging from 10 to 80 nm, especially, dominated particle number concentrations with $SO_2$ addition. The time series of total ultrafine particles (diameter < 100 nm) number concentration are shown in Fig. S2, where it is seen that $SO_2$ considerably enhanced ultrafine particle formation.

[Figure]

**Figure R1.** The total number concentrations of ultrafine particles (< 100 nm) as a function of reaction time (Exps. 1–8). The open symbols and solid symbols represent low- and high-$NO_x$ experiments, respectively.

line 323: I would be interested in more detail about the mechanics of $H_2SO_4$ as a "condensed surface for key compounds." I take this to mean reactive uptake or heterogeneous reactions of VOCs with $H_2SO_4$. However, you don't justify this assertion; I might recommend citing Wang et al. 2010 (doi.org/10.1021/es9036868) and/or Zhang et al. 2019 (doi.org/10.1021/acsearthspacechem.9b00209) for VOC heterogeneous reaction with sulfuric acid surfaces.

**Author Reply:**

The oxidation of $SO_2$ can contribute to the formation of $H_2SO_4$ in $SO_2$-involved experiments. The formed $H_2SO_4$ plays a key role in nucleation and can contribute to the increase in nucleation rate (Sipila et al., 2010; Lehtipalo et al., 2018; Yao et al., 2018), which are responsible for the short nucleation time observed in the TMB/$NO_x$/$SO_2$ regime (Wyche et al., 2009). Several $H_2SO_4$ nucleation mechanisms have been suggested, including binary nucleation, ternary nucleation, ion-induced nucleation, and so on (Sipila et al., 2010; Lehtipalo et al., 2018; Yao et al., 2018). SOA-forming compounds can be produced in the gas-phase via photooxidation of TMB, and the chamber walls and particle surfaces are competitive condensation sinks for these compounds (Charan et al., 2020). Seed particles were not introduced into the chamber during particle production experiments in this work. In the original sentence, we meant that the $H_2SO_4$-induced new particles are an attractive condensation sinks for SOA-forming compounds and can subsequently provide surfaces like seed particles onto which the compound can condense.

We have carefully read the two recommended papers. Wang et al. (2010) found that $H_2SO_4$ can effectively uptake gas-phase alkylamines, hence contributing to the growth of aerosol particles. Zhang et al. (2019) highlighted the significance of uptake of isoprene-derived epoxydiols onto sulfate aerosol particles. This process can influence the phase state, morphology, and acidity of aerosol particles. Reactive uptake or heterogeneous reactions of organic vapors with $H_2SO_4$ are important, which can promote the growth of particle and the increase in particle mass (Deng et al., 2017; Wang et al., 2010). However, it should be noted that we focused on analyzing $SO_2$ effects on the particle nucleation time in line 323 of the original text. The heterogenous

reactions promote mainly the formation of the accumulation mode particles instead of the nucleation mode particles (Lu et al., 2019). The original sentence has been revised

175    as: The formed $H_2SO_4$ could induce nucleation and increase the nucleation rate (Zhao et al., 2018; Blair et al., 2017), and these processes are responsible for the short nucleation time observed in the $TMB/NO_x/SO_2$ regime (Wyche et al., 2009).

We fully agree with the Referee that acidic surfaces on aerosol particles can
180    promote the reactive uptake of organic vapors (Zhang et al., 2019). The acid-catalyzed heterogenous reactions were already discussed in Sect. 3.1.2 in the original manuscript as follows.

"The increase in aerosol acidity could be largely responsible for the observed enhancements in SOA formation in $SO_2$-invloved experiments. The OH oxidation of
185    TMB can result in the formation of multifunctional carbonyl compounds (Liu et al., 2012; Zaytsev et al., 2019), which could promote SOA formation via acid-catalyzed heterogeneous reactions."

line 329-30: As neither this work nor that of Julin et al. attempts to explain the
190    "nonlinear dynamics of aerosol populations," rather than attributing your results to these dynamics it might be more effective to say something like "The nonlinear response of the mean particle diameter to $SO_2$ initial concentration is similar to results found by Julin et al. (2018) in a modelling study."

**Author Reply:**

195    We have carefully read the study of Julin et al. (2018) again, who suggested nonlinear response of particle size distributions to ammonia emissions. In the revised manuscript, the sentence has been revised as follows:

The nonlinear response of the particle mean diameter to $SO_2$ initial level is similar to the findings of Julin et al. (2018), who found the response of particle size distribution
200    to $NH_3$ emissions to be also nonlinear.

line 336: Is this the mean particle diameter after 30 minutes? Be explicit. You defined

initial growth rate as average for 0-30 minutes, but did not define a timeframe for mean diameter.

205 **Author reply:**

Figure R2 shows the mean diameter of aerosol particles at different reaction times, which was measured with SMPS. The particle mean diameter can stabilize after about 300 min photooxidation. Therefore, the particle mean diameter measured at 300 min in each experiment are compared in Fig. 2 in the main manuscript. The related text and

210 Fig. 2 have been updated in the revised manuscript as follows.

Page 11, lines 331–335

The particle size rapidly increased within 30 min after nucleation, and gradually reached a stable level within 300 min photooxidation (Fig. 1). Consequently, the initial growth rate ($GR_{inital}$), calculated based on the method of Kulmala et al. (2012), is defined as the

215 particle growth rate within 30 min after nucleation (Li et al., 2018). The mean diameter reported in this work represents the particle mean diameter measured at 300 min in each experiment.

[Figure]

**Figure R2.** The mean diameter of aerosol particles generated as a function of reaction

220 time.

[Figure]

**Figure 1.** Particle nucleation time as a function of initial $SO_2$ concentration under low-$NO_x$ (open circles) and high-$NO_x$ (solid circles) conditions (Exps. 1–8). The symbol color indicates the particle mean diameter and symbol size represents the particle initial growth rate. Values of the particle parameters are listed in Table S1.

line 383-84: If you believe you have not sufficiently justified the necessity of chemical composition studies, I recommend you do so in the introduction instead.

**Author Reply:**

In the Introduction, we initially highlighted the necessity to analyze the chemical composition of particles as follows:

"To better understand TMB-SOA formation and growth, detailed laboratory characterization of TMB-SOA composition and TMB oxidation mechanisms with inorganic perturbation are required."

"The mechanisms leading to secondary aerosol formation in the urban environment remain highly elusive and controversial, particularly for the processes related to changes in secondary aerosol mass and chemical composition."

However, based on the Referee's suggestion, we moved the original sentence to the Introduction to further supplement the necessity of studying particle chemical composition.

Page 2, lines 55–62

High levels of $SO_2$, $NH_3$, and VOCs have been reported in certain regions such as

Guangzhou (Zou et al., 2015), Beijing (Meng et al., 2020), Handan (Li et al., 2017) in China. During haze pollution episodes, Li et al. (2017) observed that $SO_2$ levels can be up to 200 ppb in Handan, China. A recent study also showed significant increasing $NH_3$ levels in the atmosphere over the United States and the European Union (Warner et al., 2017). However, less focus has been placed on the $SO_2$ and $NH_3$ perturbations on SOA formation and chemical composition. Aerosol particles contain a multitude of compounds with different physicochemical properties.

line 589: Figure 9 may belong in the supplement, because it has only a brief mention in the text that is limited to the observation that positive mode MS spectra skew toward lower m/z's.

**Author Reply:**

Figure 9 has been moved to the supplement.

line 594, 600: The contents of Figure 10 are also not discussed in the text, so it could be moved to the supplement as well.

**Author Reply:**

Figure 10 presents the predicted saturation mass concentrations of detected products, which is an important part of our discussion. Therefore, we keep Figure 10 in the revised manuscript.

line 509-10: use comparable measures to compare the $SO_2$-involved and $SO_2$ free conditions. I would recommend including the multiplicative factor of the change for both.

**Author Reply:**

We have modified the original sentence following the Referee's suggestion as: Increasing the $NH_3$ level to 200 ppb enhanced the particle maximum number concentration by factors of 2.0 and 1.7 under $SO_2$-free and $SO_2$-involved conditions, respectively.

line 524-26: The sentence "Initial growth rate... as shown in Fig. S6" is redundant, and to the next sentence "In SO₂-free experiments... initial growth rate of aerosol particles" I would add something like "compared to the SO₂-involved condition."

**Author Reply:**

We have deleted the sentence "Initial growth rate... as shown in Fig. S6" in the revised manuscript. In addition, we already compared the initial growth rate of particles in $SO_2$-free and $SO_2$-involved experiments in the original manuscript as: "In $SO_2$-free experiments, the increase in $NH_3$ initial concentrations led to remarkable increase of the initial growth rate of aerosol particles. High initial growth rates were also found in the photooxidation of other aromatics such as toluene and o/m/p-xylene with $NH_3$ addition (Li et al., 2018). This result may be explained by the fact that under $SO_2$-free condition, $NH_3$ mainly reacts with acids to produce ammonium salts. Previous studies have reported that ammonium salts could partition into the initial growth process of new secondary aerosol particles and thus increase the particle initial growth rate (Li et al., 2018; Zhu et al., 2014). More interestingly, $NH_3$ level did not substantially affect the initial growth rate of particles in the presence of $SO_2$ (Fig. S9)."

**References**

Blair, S. L., MacMillan, A. C., Drozd, G. T., Goldstein, A. H., Chu, R. K., Pasa-Tolic, L., Shaw, J. B., Tolic, N., Lin, P., Laskin, J., Laskin, A., and Nizkorodov, S. A.: Molecular characterization of organosulfur compounds in biodiesel and diesel fuel secondary organic aerosol, Environ. Sci. Technol., 51, 119-127, 10.1021/acs.est.6b03304, 2017.

Charan, S. M., Huang, Y., and Seinfeld, J. H.: Computational Simulation of Secondary Organic Aerosol Formation in Laboratory Chambers, Chem. Rev., 119, 11912-11944, 10.1021/acs.chemrev.9b00358, 2019.

Charan, S. M., Buenconsejo, R. S., and Seinfeld, J. H.: Secondary organic aerosol yields from the oxidation of benzyl alcohol, Atmos. Chem. Phys., 20, 13167-13190, 10.5194/acp-20-13167-2020, 2020.

Chen, T., Liu, Y., Ma, Q., Chu, B., Zhang, P., Liu, C., Liu, J., and He, H.: Significant source of secondary aerosol: formation from gasoline evaporative emissions in the presence of $SO_2$ and $NH_3$, Atmos. Chem. Phys., 19, 8063-8081, 10.5194/acp-19-8063-2019, 2019.

Crump, J. G., and Seinfeld, J. H.: turbulent deposition and gravitational sedimentation

of an aerosol in a vessel of arbitrary shape, Journal of Aerosol Science, 12, 405-415, 10.1016/0021-8502(81)90036-7, 1981.

310    Deng, W., Liu, T., Zhang, Y., Situ, S., Hu, Q., He, Q., Zhang, Z., Lu, S., Bi, X., Wang, X., Boreave, A., George, C., Ding, X., and Wang, X.: Secondary organic aerosol formation from photooxidation of toluene with $NO_x$ and $SO_2$: chamber simulation with purified air versus urban ambient air as matrix, Atmos. Environ., 150, 67-76, 10.1016/j.atmosenv.2016.11.047, 2017.

315    Ge, S., Xu, Y., and Jia, L.: Secondary organic aerosol formation from ethylene ozonolysis in the presence of sodium chloride, Journal of Aerosol Science, 106, 120-131, 10.1016/j.jaerosci.2017.01.009, 2017.

Hinks, M. L., Montoya-Aguilera, J., Ellison, L., Lin, P., Laskin, A., Laskin, J., Shiraiwa, M., Dabdub, D., and Nizkorodov, S. A.: Effect of relative humidity on the composition

320    of secondary organic aerosol from the oxidation of toluene, Atmos. Chem. Phys., 18, 1643-1652, 10.5194/acp-18-1643-2018, 2018.

Julin, J., Murphy, B. N., Patoulias, D., Fountoukis, C., Olenius, T., Pandis, S. N., and Riipinen, I.: Impacts of Future European Emission Reductions on Aerosol Particle Number Concentrations Accounting for Effects of Ammonia, Amines, and Organic

325    Species, Environ. Sci. Technol., 52, 692-700, 10.1021/acs.est.7b05122, 2018.

Kulmala, M., Petaja, T., Nieminen, T., Sipila, M., Manninen, H. E., Lehtipalo, K., Dal Maso, M., Aalto, P. P., Junninen, H., Paasonen, P., Riipinen, I., Lehtinen, K. E., Laaksonen, A., and Kerminen, V. M.: Measurement of the nucleation of atmospheric aerosol particles, Nat. Protoc., 7, 1651-1667, 10.1038/nprot.2012.091, 2012.

330    Lehtipalo, K., Yan, C., Dada, L., Bianchi, F., Xiao, M., Wagner, R., Stolzenburg, D., Ahonen, L. R., Amorim, A., Baccarini, A., Bauer, P. S., Baumgartner, B., Bergen, A., Bernhammer, A.-K., Breitenlechner, M., Brilke, S., Buchholz, A., Mazon, S. B., Chen, D., Chen, X., Dias, A., Dommen, J., Draper, D. C., Duplissy, J., Ehn, M., Finkenzeller, H., Fischer, L., Frege, C., Fuchs, C., Garmash, O., Gordon, H., Hakala, J., He, X.,

335    Heikkinen, L., Heinritzi, M., Helm, J. C., Hofbauer, V., Hoyle, C. R., Jokinen, T., Kangasluoma, J., Kerminen, V.-M., Kim, C., Kirkby, J., Kontkanen, J., Kuerten, A., Lawler, M. J., Mai, H., Mathot, S., Mauldin, R. L., III, Molteni, U., Nichman, L., Nie, W., Nieminen, T., Ojdanic, A., Onnela, A., Passananti, M., Petaja, T., Piel, F., Pospisilova, V., Quelever, L. L. J., Rissanen, M. P., Rose, C., Sarnela, N., Schallhart, S.,

340    Schuchmann, S., Sengupta, K., Simon, M., Sipila, M., Tauber, C., Tome, A., Trostl, J., Vaisanen, O., Vogel, A. L., Volkamer, R., Wagner, A. C., Wang, M., Weitz, L., Wimmer, D., Ye, P., Ylisirnio, A., Zha, Q., Carslaw, K. S., Curtius, J., Donahue, N. M., Flagan, R. C., Hansel, A., Riipinen, I., Virtanen, A., Winkler, P. M., Baltensperger, U., Kulmala, M., and Worsnop, D. R.: Multicomponent new particle formation from sulfuric acid,

345    ammonia, and biogenic vapors, Sci. Adv., 4, 10.1126/sciadv.aau5363, 2018.

Li, H., Zhang, Q., Zhang, Q., Chen, C., Wang, L., Wei, Z., Zhou, S., Parworth, C., Zheng, B., Canonaco, F., Prevot, A. S. H., Chen, P., Zhang, H., Wallington, T. J., and He, K.: Wintertime aerosol chemistry and haze evolution in an extremely polluted city of the North China Plain: significant contribution from coal and biomass combustion, Atmos.

350    Chem. Phys., 17, 4751-4768, 10.5194/acp-17-4751-2017, 2017.

Li, K., Chen, L., White, S. J., Yu, H., Wu, X., Gao, X., Azzi, M., and Cen, K.: Smog

chamber study of the role of NH3 in new particle formation from photo-oxidation of aromatic hydrocarbons, Sci. Total Environ., 619-620, 927-937, 10.1016/j.scitotenv.2017.11.180, 2018.

355     Liu, S., Shilling, J. E., Song, C., Hiranuma, N., Zaveri, R. A., and Russell, L. M.: Hydrolysis of Organonitrate Functional Groups in Aerosol Particles, Aerosol Sci. Technol., 46, 1359-1369, 10.1080/02786826.2012.716175, 2012.

Lu, K., Guo, S., Tan, Z., Wang, H., Shang, D., Liu, Y., Li, X., Wu, Z., Hu, M., and Zhang, Y.: Exploring atmospheric free-radical chemistry in China: the self-cleansing

360     capacity and the formation of secondary air pollution, National Science Review, 6, 579-594, 10.1093/nsr/nwy073, 2019.

Mehra, A., Wang, Y., Krechmer, J. E., Lambe, A., Majluf, F., Morris, M. A., Priestley, M., Bannan, T. J., Bryant, D. J., Pereira, K. L., Hamilton, J. F., Rickard, A. R., Newland, M. J., Stark, H., Croteau, P., Jayne, J. T., Worsnop, D. R., Canagaratna, M. R., Wang,

365     L., and Coe, H.: Evaluation of the chemical composition of gas- and particle-phase products of aromatic oxidation, Atmos. Chem. Phys., 20, 9783-9803, 10.5194/acp-20-9783-2020, 2020.

Meng, Z., Wu, L., Xu, X., Xu, W., Zhang, R., Jia, X., Liang, L., Miao, Y., Cheng, H., Xie, Y., He, J., and Zhong, J.: Changes in ammonia and its effects on PM2.5 chemical

370     property in three winter seasons in Beijing, China, Sci. Total Environ., 749, 142208, 10.1016/j.scitotenv.2020.142208, 2020.

Mo, Z., Lu, S., and Shao, M.: Volatile organic compound (VOC) emissions and health risk assessment in paint and coatings industry in the Yangtze River Delta, China, Environ. Pollut., 269, 115740, 10.1016/j.envpol.2020.115740, 2021.

375     Ran, L., Zhao, C., Geng, F., Tie, X., Tang, X., Peng, L., Zhou, G., Yu, Q., Xu, J., and Guenther, A.: Ozone photochemical production in urban Shanghai, China: Analysis based on ground level observations, J. Geophys. Res. Atmos., 114, 10.1029/2008jd010752, 2009.

Sipila, M., Berndt, T., Petaja, T., Brus, D., Vanhanen, J., Stratmann, F., Patokoski, J.,

380     Mauldin, R. L., III, Hyvarinen, A.-P., Lihavainen, H., and Kulmala, M.: The role of sulfuric acid in atmospheric nucleation, Science, 327, 1243-1246, 10.1126/science.1180315, 2010.

Wang, L., Lal, V., Khalizov, A. F., and Zhang, R.: Heterogeneous Chemistry of Alkylamines with Sulfuric Acid: Implications for Atmospheric Formation of

385     Alkylaminium Sulfates, Environ. Sci. Technol., 44, 2461-2465, 10.1021/es9036868, 2010.

Wang, Y., Mehra, A., Krechmer, J. E., Yang, G., Hu, X., Lu, Y., Lambe, A., Canagaratna, M., Chen, J., Worsnop, D., Coe, H., and Wang, L.: Oxygenated products formed from OH-initiated reactions of trimethylbenzene: autoxidation and accretion, Atmos. Chem.

390     Phys., 20, 9563-9579, 10.5194/acp-20-9563-2020, 2020.

Warner, J. X., Dickerson, R. R., Wei, Z., Strow, L. L., Wang, Y., and Liang, Q.: Increased atmospheric ammonia over the world's major agricultural areas detected from space, Geophys. Res. Lett., 44, 2875-2884, 10.1002/2016gl072305, 2017.

Wyche, K. P., Monks, P. S., Ellis, A. M., Cordell, R. L., Parker, A. E., Whyte, C.,

395     Metzger, A., Dommen, J., Duplissy, J., Prevot, A. S. H., Baltensperger, U., Rickard, A.

R., and Wulfert, F.: Gas phase precursors to anthropogenic secondary organic aerosol: detailed observations of 1,3,5-trimethylbenzene photooxidation, Atmos. Chem. Phys., 9, 635-665, 10.5194/acp-9-635-2009, 2009.

400 Yao, L., Garmash, O., Bianchi, F., Zheng, J., Yan, C., Kontkanen, J., Junninen, H., Mazon, S. B., Ehn, M., Paasonen, P., Sipila, M., Wang, M., Wang, X., Xiao, S., Chen, H., Lu, Y., Zhang, B., Wang, D., Fu, Q., Geng, F., Li, L., Wang, H., Qiao, L., Yang, X., Chen, J., Kerminen, V.-M., Petaja, T., Worsnop, D. R., Kulmala, M., and Wang, L.: Atmospheric new particle formation from sulfuric acid and amines in a Chinese megacity, Science, 361, 278-281, 10.1126/science.aao4839, 2018.

405 Yu, K. P., Lin, C. C., Yang, S. C., and Zhao, P.: Enhancement effect of relative humidity on the formation and regional respiratory deposition of secondary organic aerosol, J Hazard Mater, 191, 94-102, 10.1016/j.jhazmat.2011.04.042, 2011.

Zaytsev, A., Koss, A. R., Breitenlechner, M., Krechmer, J. E., Nihill, K. J., Lim, C. Y., Rowe, J. C., Cox, J. L., Moss, J., Roscioli, J. R., Canagaratna, M. R., Worsnop, D.,
410 Kroll, J. H., and Keutsch, F. N.: Mechanistic study of the formation of ring-retaining and ring-opening products from the oxidation of aromatic compounds under urban atmospheric conditions, Atmos. Chem. Phys., 19, 15117-15129, 10.5194/acp-19-15117-2019, 2019.

Zhang, Y., Chen, Y., Lei, Z., Olson, N. E., Riva, M., Koss, A. R., Zhang, Z., Gold, A.,
415 Jayne, J. T., Worsnop, D. R., Onasch, T. B., Kroll, J. H., Turpin, B. J., Ault, A. P., and Surratt, J. D.: Joint Impacts of Acidity and Viscosity on the Formation of Secondary Organic Aerosol from Isoprene Epoxydiols (IEPOX) in Phase Separated Particles, ACS Earth Space Chem., 3, 2646-2658, 10.1021/acsearthspacechem.9b00209, 2019.

Zhao, D., Schmitt, S. H., Wang, M., Acir, I.-H., Tillmann, R., Tan, Z., Novelli, A., Fuchs,
420 H., Pullinen, I., Wegener, R., Rohrer, F., Wildt, J., Kiendler-Scharr, A., Wahner, A., and Mentel, T. F.: Effects of $NO_x$ and $SO_2$ on the secondary organic aerosol formation from photooxidation of α-pinene and limonene, Atmos. Chem. Phys., 18, 1611-1628, 10.5194/acp-18-1611-2018, 2018.

Zhu, Y., Sabaliauskas, K., Liu, X., Meng, H., Gao, H., Jeong, C.-H., Evans, G. J., and
425 Yao, X.: Comparative analysis of new particle formation events in less and severely polluted urban atmosphere, Atmos. Environ., 98, 655-664, 10.1016/j.atmosenv.2014.09.043, 2014.

Zou, Y., Deng, X. J., Zhu, D., Gong, D. C., Wang, H., Li, F., Tan, H. B., Deng, T., Mai, B. R., Liu, X. T., and Wang, B. G.: Characteristics of 1 year of observational data of
430 VOCs, $NO_x$ and $O_3$ at a suburban site in Guangzhou, China, Atmos. Chem. Phys., 15, 6625-6636, 10.5194/acp-15-6625-2015, 2015.

---

## Author Comment (AC2)

We sincerely thank the Referee for the valuable comments. Our manuscript has been revised according to the comments from the Referee and our responses to the comments are as follows. For clarity, the comments are reproduced in blue, authors' responses are in black and changes in the manuscript are in red.

**Responses to Anonymous Referee #2**

General comments

The authors have studied SOA formation from photo-oxidation of 1,2,4-trimethylbenzene under different concentrations of NOx, SO2 and NH3 in an indoor smog chamber. The reactants and the particle generation and growth were monitored using a series of standard instrumentations. The chemical functional groups were characterized by ATR-FTIR in each experiment and inorganic constituents were analyzed using ion chromatography. The molecular level information is provided using UPLC-HRMS and dd-MS2 scans. Ten new organosulfates are identified in the presence of SO2, including 3 in which the origin was previously unknown and some previously reported to be originated from biogenic precursors. Formation mechanisms for 8 of the newly identified organosulfates are proposed based on previous literature. Their results indicate that SO2 is a key parameter for ultrafine particle formation. A synergistic effect of NH3 and SO2 in particle formation is also shown, indicating the importance of reducing both SO2 and NH3 emissions to improve lowering PM. Their results also suggest that ammonium sulfate form by the reaction of NH3 with H2SO4 facilitate aerosol formation and growth through condensation of organic vapors.

This article advances the current knowledge of aerosol formation from photo-oxidation of a typical aromatic hydrocarbon in the presence of NOx, SO2 and NH3, therefore is of interest to the scientific community of ACP. The manuscript is well written and organized. The experiments are well executed, methods are explained adequately, results are discussed thoroughly, and conclusions are well supported. I have some minor comments to improve the quality of this manuscript that are listed below. I recommend accepting this manuscript for publication in ACP with minor revisions.

Line 13 Indicate the major emission sources of TMB.

**Author Reply:**

Aromatic hydrocarbons can dominate the VOC budget in certain urban areas with

TMB being one of the most significant species (Ran et al., 2009). TMB is mostly emitted from anthropogenic sources such as solvent use (Mo et al., 2021). Recent studies have recognized TMB as an important species in SOA formation in the atmosphere (Wang et al., 2020; Zaytsev et al., 2019; Mehra et al., 2020). Hence, the following text has been added in the abstract.

Aromatic hydrocarbons can dominate the volatile organic compounds budget in the urban atmosphere. Among them, 1,2,4-trimethylbenzene (TMB), mainly emitted from solvent use, is one of the most important secondary organic aerosols (SOA) precursors.

Line 45 Define 'certain regions.

**Author Reply:**

The original text has been supplemented as follows: High levels of $SO_2$, $NH_3$, and VOCs have been reported in certain regions such as Guangzhou (Zou et al., 2015), Beijing (Meng et al., 2020), Handan (Li et al., 2017) in China. During haze pollution episodes, Li et al. (2017) observed that $SO_2$ levels can be up to 200 ppb in Handan,

China. A recent study also showed significant increasing $NH_3$ levels in the atmosphere over the United States and the European Union (Warner et al., 2017).

Line 54-56 Add reference/s.

**Author Reply:**

The following studies have been added to the references list.

1.  Estillore, A. D., Hettiyadura, A. P. S., Qin, Z., Leckrone, E., Wombacher, B., Humphry, T., Stone, E. A., and Grassian, V. H.: Water Uptake and Hygroscopic Growth of Organosulfate Aerosol, Environ. Sci. Technol., 50, 4259-4268, 10.1021/acs.est.5b05014, 2016.

2.  Fleming, L. T., Ali, N. N., Blair, S. L., Roveretto, M., George, C., and Nizkorodov,

S. A.: Formation of Light-Absorbing Organosulfates during Evaporation of Secondary Organic Material Extracts in the Presence of Sulfuric Acid, ACS Earth Space Chem., 3, 947-957, 10.1021/acsearthspacechem.9b00036, 2019

3.  Hansen, A. M. K., Hong, J., Raatikainen, T., Kristensen, K., Ylisirniö, A., Virtanen, A., Petäjä, T., Glasius, M., and Prisle, N. L.: Hygroscopic properties and cloud condensation nuclei activation of limonene-derived organosulfates and their mixtures with ammonium sulfate, Atmos. Chem. Phys., 15, 14071-14089, 10.5194/acp-15-14071-2015, 2015.

4.  Riva, M., Chen, Y., Zhang, Y., Lei, Z., Olson, N. E., Boyer, H. C., Narayan, S., Yee, L. D., Green, H. S., Cui, T., Zhang, Z., Baumann, K., Fort, M., Edgerton, E., Budisulistiorini, S. H., Rose, C. A., Ribeiro, I. O., RL, E. O., Dos Santos, E. O., Machado, C. M. D., Szopa, S., Zhao, Y., Alves, E. G., de Sa, S. S., Hu, W., Knipping, E. M., Shaw, S. L., Duvoisin Junior, S., de Souza, R. A. F., Palm, B. B., Jimenez, J. L., Glasius, M., Goldstein, A. H., Pye, H. O. T., Gold, A., Turpin, B. J., Vizuete, W., Martin, S. T., Thornton, J. A., Dutcher, C. S., Ault, A. P., and Surratt, J. D.: Increasing Isoprene Epoxydiol-to-Inorganic Sulfate Aerosol Ratio Results in Extensive Conversion of Inorganic Sulfate to Organosulfur Forms: Implications for Aerosol Physicochemical Properties, Environ. Sci. Technol., 53, 8682-8694, 10.1021/acs.est.9b01019, 2019.

5.  Zhang, Y., Chen, Y., Lei, Z., Olson, N. E., Riva, M., Koss, A. R., Zhang, Z., Gold, A., Jayne, J. T., Worsnop, D. R., Onasch, T. B., Kroll, J. H., Turpin, B. J., Ault, A. P., and Surratt, J. D.: Joint Impacts of Acidity and Viscosity on the Formation of Secondary Organic Aerosol from Isoprene Epoxydiols (IEPOX) in Phase Separated Particles, ACS Earth Space Chem., 3, 2646-2658, 10.1021/acsearthspacechem.9b00209, 2019.

They are cited in the revised manuscript as follows:

The presence of OSs could alter aerosol morphology (Riva et al., 2019), viscosity (Riva et al., 2019; Zhang et al., 2019), particle acidity (Riva et al., 2019), phase state (Zhang et al., 2019), hygroscopicity (Estillore et al., 2016; Hansen et al., 2015), and optical properties (Fleming et al., 2019), thereby resulting in large climate effects.

Line 98-99 Briefly indicate what MCM is and add a link to the version.

**Author Reply:**

The Master Chemical Mechanism (MCM, http://mcm.leeds.ac.uk/MCMv3.2/, last access: 23 February 2021) is a near-explicit chemical mechanism that can describe, in detail, the tropospheric degradation of numerous VOCs. The oxidation processes of VOCs are highly complex. Explicit chemical mechanisms might cause missing oxidation pathways and products, thus leading to large gaps between the modeled and measured SOA mass. To the best of our knowledge, the oxidation mechanism of TMB has not been updated in the MCM since 2005. Wang et al. (2020) suggested that recently identified autoxidation pathways for OH oxidation of TMB were not included in the current MCM and the detected products are more diverse than the products shown in MCM. The updates for the OH-initiated oxidation mechanism of TMB can be achieved only when the rate, branching ratios and product distributions can be explicitly obtained. Therefore, more investigations are needed to further explore the detailed chemical processes for OH oxidation of TMB. According to the suggestion of the Referee, the following text was added in the revised manuscript.

Page 4, lines 124–134

In addition, the Master Chemical Mechanism (MCM, http://mcm.leeds.ac.uk/MCMv3.2/, last access: 23 February 2021) is a near-explicit chemical mechanism that can describe, in detail, the tropospheric degradation of numerous VOCs. A recent study reported that identified autoxidation pathways during OH oxidation of TMB were not included in the current MCM and the detected TMB products were more diverse than the products shown in MCM (Wang et al., 2020). TMB photooxidation is highly complex and sensitive to environmental conditions. The updates for the OH-initiated oxidation mechanism of TMB can be achieved only when the rate constants, branching ratios and product distributions can be explicitly obtained. It is necessary to further investigate the detailed chemical processes for TMB photooxidation.

Line 210 Indicate how the mass calibration was performed.

**Author Reply:**

We used standard calibration solutions provided by the manufacturer to calibrate the MS instrument every five days. The following text was added in the revised manuscript.

Page 9, lines 259–260

The MS instrument was calibrated every five days with standard calibration solutions provided by the manufacturer.

Line 200 Indicate the mass resolution in full MS, top N in dd-MS2, isolation width (mass window) etc

**Author Reply:**

In the revised manuscript, detailed parameters are indicated as follows:

Page 9, lines 253–259

MS spectra were recorded in the range of $m/z$ 50 to 750 in full MS scan with a mass resolving power of 70000 (FWHM at $m/z$ 200). The full MS scan was followed by data-dependent MS/MS (dd-MS$^2$) scans using stepped collision energies of 20, 40, and 60 eV via high-energy collisional dissociation. The resolution was 17500 and an isolation width of 2 $m/z$ units was applied for the dd-MS$^2$ scan. The other parameters for MS$^2$ experiments were as follow: AGC target, $2 \times 10^5$; maximum IT, 50 ms; loop count, 3; minimum AGC target, $1 \times 10^5$; apex trigger, 2–6 s; dynamic exclusion, 6 s.

Line 277 Add initial growth rates to Table 1 or SI.

**Author Reply:**

The aim of depicting Fig. 2 in the original manuscript was to clearly reflect the changes of the particle nucleation time, initial growth rate, and mean diameter with the initial $SO_2$ concentration. The specific particle parameters including the particle initial growth rate have been summarized in the table below.

**Table R1.** Particle parameters for experiments 1–8.

| Exp. | Nucleation time (min) | Particle mean diameter (nm) | Initial particle growth rate (nm h$^{-1}$) |
|------|------|------|------|
| 1 | 70 | 125.5 | 46.53 |
| 2 | 15 | 109.9 | 20.09 |
| 3 | 10 | 121.2 | 27.42 |
| 4 | 10 | 130.6 | 31.09 |
| 5 | 100 | 123.1 | 23.51 |
| 6 | 20 | 118.5 | 19.30 |
| 7 | 10 | 112.4 | 22.14 |
| 8 | 10 | 136.5 | 29.82 |

Table R1 has been added in the revised supplement as Table S1 and the caption of Fig. 2 have been also updated as follows.

**Figure 2.** Particle nucleation time as a function of initial SO$_2$ concentration under low-NO$_x$ (open circles) and high-NO$_x$ (solid circles) conditions (Exps. 1–8). The symbol color indicates the particle mean diameter and symbol size represents the particle initial growth rate. Values of the particle parameters are listed in Table S1.

Line 277-280 Can the authors elaborate the reasons for the observed non-linear response of the particle diameter with initial SO$_2$ concentrations?

**Author Reply:**

OH-initiated oxidation of VOCs can yield a number of products with different degrees of volatility (Xu et al., 2014). The growth of aerosol particles is related to the coagulation, the condensation of non-volatility oxidation products, and gas-particle partitioning of semi-volatility organic compounds (SVOCs). For SVOCs, the evaporation is important after partitioning to the particle phase. Therefore, the rate at which SVOCs participate in the particle growth is lower than their condensation rate. Interestingly, recent studies showed that the particle-phase chemistry such as heterogeneous reactions of SVOCs are substantially pronounced for particle growth (Shiraiwa et al., 2013; Paasonen et al., 2018; Apsokardu and Johnston, 2018). Apsokardu and Johnston (2018) explored the influences of particle-phase chemistry on the growth rate of aerosol particles with a kinetic growth model and found that some

SVOCs can undergo accretion reactions in the particle-phase to further accelerate the particle growth. Paasonen et al. (2018) also showed that the increase in particle growth rate is related to particle-phase reactions. The study of Shiraiwa et al. (2013) highlighted the importance of particle-phase chemistry in the changes of SOA size distribution and suggested that particle-phase chemistry depends to some extent on particle acidity. Organosulfates can be produced by particle-phase reactions involving interactions between organics and inorganics. In this work, we measured organosulfates only in $SO_2$-involved photooxidation, indicating that additional particle-phase reactions can occur under $SO_2$-involved conditions. Increasing initial $SO_2$ levels could induce the formation of more sulfate (Fig. S4) and the enhancement in the particle acidity during photooxidation (Liu et al., 2016; Kroll et al., 2006). The elevated particle acidity can promote more SVOCs to transform into non-volatile products in the particle phase (Lin et al., 2014; Lal et al., 2012). Additional SVOCs could be moved from the gas phase to particle phase to increase the particle size and the evaporation of SVOCs can be inhibited, thereby promoting the particle growth.

It is somewhat interesting that the particle mean diameter in low-$SO_2$ ($[SO_2]_0 < 100$ ppb) experiments is smaller than that in $SO_2$-free experiments. Our result is in line with the study of Wyche et al. (2009), who attributed this phenomenon to the larger number of particles produced under $SO_2$-involved condition. In this work, the presence of 59 ppb $SO_2$ caused the maximum number concentration of particles to increase by $8.5 \times 10^4$ $cm^{-3}$ under low-$NO_x$ condition. When the $SO_2$ level increased from 0 to 68 ppb in high-$NO_x$ experiments, the corresponding particle number concentration increased from $2.9 \times 10^4$ to $9.3 \times 10^4$ $cm^{-3}$. Therefore, the amounts of products that condensed onto each aerosol particles may significantly decrease in low-$SO_2$ experiments, which can result in the decrease in particle diameter (Liu et al., 2015). The promoting effect of particle-phase chemistry on the particle size growth may not offset the inhibiting effect of the emergence of large number of particles on the particle size growth, thereby leading to the low particle diameter in low-$SO_2$ experiments. We have added the following text to explain the reasons for the observed non-linear response of the particle diameter with initial $SO_2$ level.

Aerosol particles can grow in different ways such as gas-particle partitioning of semi-volatility organic compounds (SVOCs). Since the evaporation of SVOCs is important after partitioning to the particle phase, the rate at which SVOCs participate in the particle growth is lower than their condensation rate. However, recent advances give an insight that the particle-phase chemistry such as heterogeneous reactions of SVOCs are substantially pronounced for the particle growth (Shiraiwa et al., 2013; Paasonen et al., 2018; Apsokardu and Johnston, 2018). Organosulfates can be produced by particle-phase reactions involving interactions between organics and inorganics. In this work, organosulfates were only detected in $SO_2$-involved photooxidation, indicating that additional particle-phase reactions can occur under $SO_2$-involved conditions. Increasing the initial $SO_2$ level could induce the formation of more sulfate (Fig. S4) and the enhancement in the particle acidity during photooxidation (Liu et al., 2016; Kroll et al., 2006). The elevated particle acidity can promote more SVOCs to transform into non-volatile products such as oligomers and other high molecular mass compounds in the particle phase, thereby promoting the particle growth (Lin et al., 2014; Lal et al., 2012). Then, additional SVOCs could be transferred from the gas phase to the particle phase to increase the particle size. However, the particle mean diameter in low-$SO_2$ ([$SO_2$]$_0$ < 100 ppb) experiments is smaller than that in $SO_2$-free experiments. Our result is in line with the study of Wyche et al. (2009), who attributed this phenomenon to the larger number of particles produced under $SO_2$-involved condition. The presence of 59 ppb $SO_2$ caused the maximum number concentration of particles to increase by $8.5 \times 10^4$ cm$^{-3}$ under low-$NO_x$ condition. When the $SO_2$ level increased from 0 to 68 ppb in high-$NO_x$ experiments, the corresponding particle number concentration increased from $2.9 \times 10^4$ to $9.3 \times 10^4$ cm$^{-3}$. Therefore, the amounts of products that condensed onto each aerosol particles significantly decreased in low-$SO_2$ experiments, which could result in the decrease in particle diameter (Liu et al., 2015). The promoting effect of particle-phase chemistry on the particle size growth may not offset the inhibiting effect of the emergence of large number of particles on the particle size growth, thereby leading to the low particle diameter in low-$SO_2$ experiments.

**Author Reply:**

The error bars represent uncertainties in the SOA yield results. The uncertainties were calculated from error propagation using the sum of the uncertainties in TMB data and the systematic error of SMPS. The following note was included in the caption of Fig. 4.

The error bars represent uncertainties in the SOA yield results and the uncertainties 240    were calculated from error propagation using the sum of the uncertainties in TMB data and the systematic error of SMPS.

Figure 5 – Indicate the experiment numbers relevant to a) to f)

**Author Reply:**

The experimental numbers have been added in the caption of Fig. 5 as:

**Figure 5.** ATR-FTIR spectra of aerosol particles generated from $TMB/NO_x$ (a, Exp. 1; b, Exp. 5), $TMB/NO_x/SO_2$ (c, Exp. 4; d, Exp. 8), $TMB/NO_x/NH_3/SO_2$ (e, Exp. 12), and $TMB/NO_x/NH_3$ (f, Exp. 10) photooxidation.

Lines 346-348 – Add detailed information of the common products observed in SO2 free and SO2 involved experiments to the SI.

**Author Reply:**

The detailed information of representative products mentioned in the original manuscript were provided in Table S3 in the revised supplement.

Table S3 – Indicate clearly whether this table shows the compounds that are detected in both SO2 free and SO2 involved experiments with NH3 or the products only formed from the experiments involved both NH3 and SO2.

**Author Reply:**

The major products shown in Table S3 are the products detected in both $SO_2$-free and $SO_2$-involved experiments with $NH_3$ addition. The revised table in the supplement is now labeled as Table S4 and the caption was revised as: Observed products in both $SO_2$-free and $SO_2$-involved experiments with $NH_3$ addition.

Tables S2 and S3 – Add UPLC retention times.

**Author Reply:**

The retention times have been added in the indicated tables, which are now labelled as Table S3 and S4.

Technical corrections

Line 105 – Add 'in the atmosphere' to the end of "Given the ubiquity of SO2, NH3, and TMB…

**Author Reply:**

According to the suggestion of the Referee, we have added "in the atmosphere" at the indicated place in the revised manuscript.

Line 200 – It is better to write it as data-dependent MS/MS (dd-MS2) scans

**Author Reply:**

We have modified it in the text.

Line 206 – Add B after 3%.

**Author Reply:**

We have added "B" in the revised manuscript.

Figure 6 – Label the red structures as OS-226, OS-228…etc. (Authors may replace the chemical formula with their abbreviated names as the structures are shown.)

**Author Reply:**

The red structures in Fig. 6 have been marked with OS-266, OS-228…etc.

Figure 10 – Match the color of the TMB on the figure with carbon number (should be light blue as it has 9 carbons)

**Author Reply:**

We have corrected it in the revised manuscript.

**References**

[revised manuscript text omitted]

---

## Author Comment (AC3)

We sincerely thank the Referee for the valuable comments. Our manuscript has been revised according to the comments from the Referee and our responses to the comments are as follows. For clarity, the comments are reproduced in blue, authors' responses are in black and changes in the manuscript are in red.

**Responses to Anonymous Referee #3**

**Overview:**

The manuscript by Yang et al. examined the effect of $SO_2$ and $NH_3$ on the formation of SOA from 1,2,4-trimethylbenzene photooxidation. After the injection of $SO_2$ and/or $NH_3$, the apparent yield of the SOA increased after wall loss correction, which demonstrated the synergistic effect of $SO_2$ and $NH_3$ in facilitating SOA formation. The authors also used ATR-FTIR, IC, and UPLC-MS to systematically analyze the particle phase composition and identified various inorganic and organo-sulfates compounds. My main comments are about the control experiment setup in this manuscript, which are discussed below. Other than the main comments, the manuscript is written clearly and comprehensive.

**Major Comments:**

The authors demonstrate that the addition of $SO_2$ and/or $NH_3$ during the photo-oxidation experiments can enhance the SOA formation and yield. However, part of the aerosol growth can arise from the formation of $H_2SO_4$ or $(NH_4)_2SO_4$ particles even without the presence of any SOA, which should be deducted from the enhancement effect of $SO_2$ and/or $NH_3$. I suggest the authors add in three control experiments (pure $SO_2$, pure $NH_3$, and $SO_2+NH_3$ mixtures but with similar levels of OH radicals and UV intensity using $H_2O_2$ or other OH generators) without any VOCs to rule out the formation of inorganic species contributing to the SOA.

**Author Reply:**

In the $SO_2$-added experiments, $H_2SO_4$ can be produced by OH oxidation of $SO_2$ upon photooxidation. With the coexistence of $SO_2$ and $NH_3$, the formed $H_2SO_4$ can be neutralized by $NH_3$ to form $(NH)_4SO_4$ particles. Therefore, the addition of $SO_2$ and/or $NH_3$ to the TMB/$NO_x$ mixtures can lead to the increase in the volume concentration of aerosol particles by the formation of inorganic species and/or the enhanced organic products (Ye et al., 2018; Jaoui et al., 2008; Liu et al., 2019). In the present study, we discussed the $SO_2$ dependence of the SOA yield as shown in Fig. 4, and we showed the role of $NH_3$ in the total particle number and volume concentrations (as discussed in Sect. 3.2.1). We believe that the introduction of $SO_2$ can enhance SOA yield based on our careful analysis as following:

(1) Some studies on the effects of $SO_2$ and/or $NH_3$ on SOA formation employed aerosol mass spectrometer to measure in-situ the inorganic and organic components in aerosol particles (Liu et al., 2016; Chen et al., 2019). For our online measurement of aerosol particles, we solely used SMPS, whose data are insufficient to quantitatively explain the contribution of organic aerosols to the increase in aerosol mass. Offline measurement methods are also commonly adopted in chamber studies. Ion chromatography (IC) is one of the most widely used instruments providing inorganic aerosol information. Kleindienst et al. (2006) investigated the $SO_2$ effects on SOA yields from the isoprene/$NO_x$/$SO_2$ photooxidation and used IC to determine the mass concentrations of inorganic components. Jiang et al. (2020) revealed the influences of $SO_2$ and $NH_3$ on furan SOA yield based on SMPS and IC measurements. In this work, the concentrations of inorganic ions were measured through IC. To determine the net

SOA yield, the mass of inorganic components was subtracted from the total particle mass based on IC and SMPS data as mentioned in Sect. 3.1.2 in the original manuscript (Jiang et al., 2020). When $SO_2$ initial levels increased from 0 to 200 ppb, the net SOA yield increased from 3.8% to 17.6% in the low-$NO_x$ regime. Similarly, elevating $SO_2$ initial concentration to 228 ppb under high-$NO_x$ condition enhanced the net SOA yield by a factor of 3.49. The promoting effects of $SO_2$ on SOA yields were in line with previous studies (Chen et al., 2019; Liu et al., 2019).

(2) We used an estimation method to explore the contribution of the generated $H_2SO_4$ to the particle formation enhancement in TMB/$NO_x$/$SO_2$ photooxidation, where we assumed the full conversion of the consumed $SO_2$ into $H_2SO_4$ aerosol particles (Ye et al., 2018; Wyche et al., 2009). As shown in Fig. R1, the contribution of the formed $H_2SO_4$ to the increase in particle volume concentration was less than 100%. Furthermore, a previous study showed that half of the reacted $SO_2$ could transform into sulfur-containing organic species during the photooxidation of 1,3,5-trimethylbenzene/o-xylene/octane/toluene (Vivanco et al., 2011). HRMS

measurements reveal the OSs production in this work, which may result in the decrease in the amount of $H_2SO_4$ in particle phase. Therefore, the enhanced SOA formation is also responsible for the increased particle volume concentration in the presence of $SO_2$.

[Figure]

**Figure R1.** Contribution (%) of the formed $H_2SO_4$ to the increase in particle volume concentration during low-$NO_x$ and high-$NO_x$ experiments.

(3) We also performed different experiments without introducing TMB, which could provide significant information about secondary inorganic aerosol formation as suggested by the Referee. In TMB/$NO_x$/$SO_2$ photooxidation, the consumption of 9.9

and 23.3 ppb $SO_2$ can cause the particle volume concentration to increase by 32.9 and 89.2 $\mu m^3$ $cm^{-3}$, respectively. In pure $SO_2$ photooxidation, the volume concentrations of the formed particles were only 25.3 and 43.2 $\mu m^3$ $cm^{-3}$ when the consumptions of $SO_2$ were 9.5 and 24.2 ppb, respectively. Comparison of the results of TMB/$NO_x$/$SO_2$ and pure SO$_2$ oxidation experiments demonstrates that the enhancement in aerosol particles by SO$_2$ introduction cannot be solely attributed to inorganic aerosol formation.

The yields shown in Table 1, obtained after ruling out the influences of inorganic species, were net SOA yields. Now, the following note was included in Table 1 in the revised manuscript. For SOA mass calculation, the inorganic mass concentration has been subtracted from the particle mass concentration.

Besides, we have added the following text in the revised manuscript and supplement to provide more evidence about SO$_2$ effects on the net SOA yield.

Page 14, lines 401–407 in the main manuscript

We assumed full conversion of the consumed SO$_2$ into H$_2$SO$_4$ aerosol particles and found that the contribution of the formed H$_2$SO$_4$ to the increase in particle volume concentration was less than 100% (See Sect. S2). In addition, pure SO$_2$ oxidation experiments without TMB addition also indicated that the enhancement in aerosol particles by SO$_2$ introduction cannot be solely attributed to inorganic aerosol formation (See Sect. S2). To calculate the net SOA yield, the inorganic mass concentration was subtracted from the particle mass concentration based on IC measurements of generated particles.

Supplement

**S2. The formed H$_2$SO$_4$ estimation and inorganic mixture experiments**

In order to evaluate the SO$_2$ effects on SOA formation, we used the method of Ye et al. (2018) to calculate the contribution of the generated H$_2$SO$_4$ to the particle formation enhancement in TMB/NO$_x$/SO$_2$ photooxidation (Ye et al., 2018; Wyche et al., 2009), where we assumed full conversion of the consumed SO$_2$ into H$_2$SO$_4$ aerosol particles. The contribution of the formed H$_2$SO$_4$ to the increase in particle volume concentration was less than 100% (Fig. S6), demonstrating that the enhanced SOA formation is also responsible for the increased particle volume concentration. Additionally, a previous study has shown that half of the reacted SO$_2$ can transform into sulfur-containing organic species during the photooxidation of 1,3,5- trimethylbenzene/o-xylene/octane/toluene (Vivanco et al., 2011). HRMS measurements revealed the OSs production in this work, which may result in the decrease in the amount of $H_2SO_4$ in the particle phase. Therefore, the enhancement in aerosol particles by $SO_2$ introduction cannot be solely attributed to inorganic aerosol formation. Pure $SO_2$ oxidation experiments without introducing TMB were also carried out. In the TMB/$NO_x$/$SO_2$ regime, the consumption of 9.9 and 23.3 ppb $SO_2$ could cause the particle volume concentration to increase by 32.9 and 89.2 $\mu m^3\ cm^{-3}$, respectively. However, in pure $SO_2$ oxidation experiments, the volume concentrations of the formed particles were only 25.3 and 43.2 $\mu m^3\ cm^{-3}$ when the consumptions of $SO_2$ were 9.5 and 24.2 ppb, respectively. Comparison of the results of TMB/$NO_x$/$SO_2$ and pure $SO_2$ oxidation experiments demonstrates that the enhancement in aerosol particles by $SO_2$ introduction cannot be solely attributed to inorganic aerosol formation.

[Figure]

**Figure S6.** Contribution (%) of the formed $H_2SO_4$ to the increased particle volume concentration during low-$NO_x$ and high-$NO_x$ experiments.

Figure S7 shows the mass spectra with and without $SO_2$ and/or $NH_3$ are similar, suggesting that maybe inorganic species formed with OH are the likely source of enhancement. It would be important to understand that after ruling out this part of the inorganic aerosol formation, how much enhancement the SO₂ and/or NH₃ would add to the yield of the SOA.

**Author Reply:**

It should be noted that Fig. S7 (now labeled as Fig. S10 in the revised supplement) compared the mass spectra of aerosol particles from NH$_3$-free and NH$_3$-added experiments. First, we carried out pure NH$_3$ oxidation experiments and found that aerosol particles were not formed within 480 min of UV irradiation. In addition, the promoting effect of NH$_3$ on SOA formation was not as strong as that of SO$_2$ with similar level. Under SO$_2$-free condition, the net SOA yield increased slightly from 3.5% to 5.1% as NH$_3$ initial level increased from 0 to 200 ppb. Our result is consistent with the finding of Chen et al. (2020a), who showed that NH$_3$ did not significantly affect SOA formation from toluene/NO$_x$ photooxidation under dry condition. Therefore, the NH$_3$-induced changes in the absolute concentrations of organic components might be small in SO$_2$-free experiments, leading to similar mass spectra for Fig. S10(a) and Fig. S10(b). Under SO$_2$-involved conditions, the introduction of NH$_3$ resulted in MS differences in the range of $m/z$ 200–400 as presented in Fig. S10(d). The sum of the ion signals in the ranges of 20–199, 200−299, 300−399, and 400−750 were compared in Fig. S11, where it can be seen that the abundance of organic compounds with $m/z > 200$ were enhanced with the addition of NH$_3$ in TMB/NO$_x$/SO$_2$ photooxidation.

We have revised the text to explain why the mass spectra of aerosol particles from NH$_3$-free and NH$_3$-added experiments are similar.

Page 22, lines 640–645

As shown in the MS spectra of aerosol samples (Fig. S10), under SO$_2$-free condition, the presence of NH$_3$ did not result in considerable changes in peak numbers and abundance for both positive ion mode and negative ion mode. NH$_3$ could slightly enhance SOA formation in SO$_2$-free experiment as mentioned in Sect. 3.2.1. Therefore, the NH$_3$-induced changes in the absolute concentrations of organic components might be small in SO$_2$-free experiments, leading to similar mass spectra for Fig. S10(a) and Fig. S10(b). In addition, the major products (Table S4) are likely generated by similar chemical mechanisms (Fig. 8), which are not sensitive to the change in initial $NH_3$ levels under current experimental conditions.

It is also necessary to show how much enhancement $NH_3$ would add to the net SOA yield according to the Referee's suggestion. Hence, the following discussions were added in the revised manuscript.

Page 20, lines 565–576

However, the effect of $NH_3$ on particle formation was not as pronounced as that of $SO_2$ with similar concentration (Fig. 7). In TMB/$NO_x$/$NH_3$ photooxidation, the net SOA yield increased slightly from 3.5% to 5.1% as $NH_3$ initial level increased from 0 to 200 ppb (Table 1). Our result is consistent with the finding of Chen et al. (2020a), who showed that $NH_3$ did not significantly affect SOA formation from toluene/$NO_x$ photooxidation under dry condition. Interestingly, SMPS measurements demonstrated that the coexistence of $SO_2$ and $NH_3$ can considerably promote secondary aerosol formation (Fig. 7). After subtracting the inorganic components, it was seen that the net SOA yield could increase to 13.7% with the introduction of 200 ppb $NH_3$ and 234 ppb $SO_2$, indicating the synergetic effects of $NH_3$ and $SO_2$ (Chu et al., 2016).

Another comment I have is that the enhancement of the SOA from the addition of $SO_2$ and/or $NH_3$ can also be attributed to the increase of the surface area from the formation of inorganic species, which shifts the gas-particle equilibrium more to the particle side. Can the authors discuss more about the effect of this shift of equilibrium? It seems Figure 4 shows that the first three experiments after adding $SO_2$ seem to follow this rule.

**Author Reply:**

An important mechanism of SOA formation and growth is gas-particle partitioning of semi-volatile compounds (SVOCs) generated from the atmospheric oxidation of VOCs. The gas-particle partitioning of SVOCs have a great sensitivity to particle surface areas in the batch-mode chamber experiments (Zhang et al., 2015; Han et al., 2019). Increasing particle surface area can limit the gas-wall interactions of organic vapors and is favorable for moving more SVOCs from the gas phase to the particle side (Han et al., 2019). These additional SVOCs may undergo further particle chemistry to strongly enhance aerosol particle formation (Apsokardu and Johnston, 2018).

Recently, the effects of the particle surface area concentration on organic aerosol formation have been explored by Han et al. (2019), who found that increasing the particle surface area concentrations can significantly increase organic aerosol mass yield due to greater partitioning of semi-volatility organic products to the particle-phase. In addition, some studies have confirmed that the SOA yields depend on the surface areas of inorganic aerosols when condensation of organic vapors onto particles is kinetically limited (McVay et al., 2014; Nah et al., 2016; Zhang et al., 2014). In the present work, the surface area concentrations of aerosol particles increased with increasing mixing ratios of $SO_2$ and/or $NH_3$ inside the chamber (Table 1), which may facilitate the gas-particle equilibrium shifting to the particle phase. The particle surface area concentrations have been included in Table 1. Based on the suggestion of the Referee, we have enriched the discussion in the revised manuscript as following:

**3.1.2 SOA yield in $SO_2$-added photooxidation**

Page 14, lines 422–438

In addition, the particle surface area concentrations significantly increased with increasing $SO_2$ initial concentrations in both low-$NO_x$ and high-$NO_x$ conditions (Table 1), which might also result in the enhancement in the SOA yield. Besides gas-particle partitioning of SVOCs, the fate of SVOCs in the chamber also include chemical reactions and chamber wall losses. Therefore, in the batch-mode chamber experiments, the gas-particle partitioning of SVOCs have a great sensitivity to particle surface areas (Zhang et al., 2015; Han et al., 2019). Recently, Zhao et al. (2018) examined the $SO_2$ effects on the SOA formation and suggested that providing additional particle surfaces by $SO_2$-induced new particle formation leads to the increase in SOA yield. The effects of the particle surface area concentration on organic aerosol formation were explored by Han et al. (2019), who also found that increasing the particle surface area concentrations can significantly increase the organic aerosol mass yield due to greater partitioning of semi-volatility organic products to the particle-phase. Increasing the particle surface area can limit the gas-wall interactions of organic vapors and is favorable for the movement of more SVOCs from the gas phase to the particle side (Han et al., 2019). These additional SVOCs can also undergo further particle chemistry such as acid-catalyzed heterogenous reactions to strongly enhance aerosol particle formation in TMB/NO$_x$/SO$_2$ photooxidation (Apsokardu and Johnston, 2018).

**3.2.1 Particle formation and growth in NH$_3$-involved photooxidation**

Page 20, lines 576–580

The flux of the gas-phase products diffusing to a particle partly depends on the surface area of the particle. The coexistence of SO$_2$ and NH$_3$ promoted the increase in particle surface area concentrations (Table 1). The ability of particle formation originating from gas-to-particle conversion may be significantly stronger with SO$_2$ and NH$_3$ introduction, leading to the enhancement in particle formation.

**3.2.2 Particle chemical composition in NH$_3$-involved photooxidation**

Page 23, lines 665–668

The introduction of SO$_2$ and NH$_3$ lead to the formation of ammonium sulfate (Fig. S12), which is an attractive condensation sink for organic vapors. High particle surface area concentration in TMB/NO$_x$/SO$_2$/NH$_3$ experiments may increase the abundance of organic compounds in the bulk phase.

**Minor Comment:**

L217: Can the author make an additional plot in the SI or show how the size dependent wall loss factor is generated?

**Author Reply:**

The wall-loss of particles are commonly evaluated by seed-only experiments where inert ammonium sulfate (AS) particles are used (Chen et al., 2019; Charan et al., 2020). In the current work, the AS solution was added to a TSI atomizer (Model 3076) to produce droplets, which passed simultaneously through a silica gel diffusion dryer to inject dry AS particles into the chamber. AS particles were lost to the chamber walls due to diffusion, gravitational settling, and electrostatic forces during experiment and the mass size distributions of AS particles were measured by SMPS for 480 min.

The size-dependent particle wall-loss rate constants were determined based on the SMPS-measured particle size distribution. First-order loss rate constants ($k_i$) of particles in each size bin $i$ across all measured sizes were firstly calculated as the slope of the corresponding ln-linear fit line:

$$\ln[M_i(t)] = -k_i t + C \qquad \text{(R1)}$$

where $M_i$ ($\mu g\ m^{-3}$) is the mass concentration of particles in the size bin $i$ at time $t$ (min)

and $C$ is an arbitrary constant. Then, the relationship between the $k_i$ and the particle diameter ($d_{p,i}$) can be described as follows:

$$k_i(d_{p,i}) = ad_{p,i}^{b} + cd_{p,i}^{-d} \qquad \text{(R2)}$$

The optimized fitted line shown in Fig. R2 can express well our independent seed experimental results.

[Figure]

**Figure R2.** Wall loss rate constant of particles as a function of particle diameter.

The description about the wall loss experiments has been updated in the revised manuscript as follows:

In order to determine the particles wall loss rates, we carried out independent wall loss experiments. An aqueous solution of ammonium sulfate was fed to a constant output atomizer (Model 3706, TSI, USA) to produce droplets, which passed simultaneously through a silica gel diffusion dryer to introduce dry particles into the chamber. The size distributions of ammonium sulfate particles were measured by SMPS for 480 min. The wall losses of particles are size-dependent and, thus, we used a size-dependent particle wall-loss correction approach, which is described in detail in the supplement.

Besides, the following text and Fig. R2 were added in the revised supplement to explain how the size dependent wall loss constant was obtained.

**S1. Size-dependent wall loss correction method**

In the present work, the size-dependent particle wall-loss rate constants were determined based on the SMPS-measured particle size distribution. The first-order loss rate constants ($k_i$) of particles in each size bin $i$ across all measured sizes were firstly calculated as the slope of the corresponding ln-linear fit line:

$$\ln[M_i(t)] = -k_i t + C \qquad \text{(S1)}$$

where $M_i$ ($\mu g\ m^{-3}$) is the mass concentration of particles in size bin $i$ at time $t$ (min) and $C$ is an arbitrary constant. Then, the relationship between the $k_i$ and the particle diameter ($d_{p,i}$) can be described as follows:

$$k_i(d_{p,i}) = a d_{p,i}^{b} + c d_{p,i}^{-d} \qquad \text{(S2)}$$

The optimized fitted line shown in Fig. S1 can express well our independent seed experimental results. Parameters a, b, c, and d in Eq. (S2) were determined to be $5.5 \times 10^{-6}$, 1.05, 0.18, 1.19, respectively. Therefore, the size-dependent loss rate ($k$) of ammonium sulfate particles can be expressed as $k = 5.5 \times 10^{-6} \times d_p^{1.05} + 0.18 \times d_p^{-1.19}$.

[Figure]

**Figure S1.** Wall loss rate constant of particles as a function of particle diameter.

Figure 1: It would be better to change the plots all in the same scale for easy comparison of different conditions.

**Author Reply:**

Figure 1 has been revised according to the comment of the Referee as follows:

[Figure]

**Figure 1.** Evolutions of the number distributions of aerosol particles generated from TMB photooxidation in low-$NO_x$ (Panels a–d) and high-$NO_x$ (Panels e–h) experiments.

**Author Reply:**

The extra space has been deleted in the revised manuscript.

L367: Please add the recent paper by Chen et al. 2020 also demonstrate that OS-228 could be from isoprene-derived SOA.

**Author Reply:**

We have cited the study of Chen et al. (2020b) and added the related description. Please, refer to the last comment.

L389: The author should also consider adding Zhang et al. 2019 here, which discussed the effects of inorganic sulfates to organosulfates conversion in affecting aerosol growth, multiphase chemistry, and pH.

**Author Reply:**

The paper by Zhang et al. (2019) has now been cited and the original sentence was modified as follow:

The conversion of inorganic sulfates to organosulfates could cause changes in aerosol growth, multiphase chemistry, and acidity (Zhang et al., 2019; Riva et al., 2019).

310 L525: As mentioned, OS-228 was observed by Chen et al. 2020 from the aging of isoprene-SOA. Even though the source might be different from the OS-228 here, the author should mention it in the conclusion instead of defining OS-228 as from unknown source.

**Author Reply:**

We have carefully read the literature suggested by the Referee. Chen et al. (2020b) investigated the heterogeneous OH oxidation of 2-methyltetrol sulfate diastereomers and identified OS at $m/z$ 227 as the oxidation product of 2-methyltetrol sulfate diastereomers. Methyltetrol sulfates are significant tracers for isoprene-derived SOA. Therefore, the OH aging of isoprene SOA is also a potential source of OS-228 in the atmosphere. The inappropriate conclusion has been removed in the revised manuscript. We have added the following text in Sect. 3.1.3 to describe the findings in the study by Chen et al. (2020b).

Page 18, lines 507–512

More recently, Chen et al. (2020b) suggested that heterogeneous OH oxidation of isoprene-derived SOA can contribute to the formation of an organosulfate with molecular weight at 228. Our results show the detection of OS-226, OS-228, OS-240, and OS-268 organosulfates, which are isomers of organosulfates derived from isoprene (Cai et al., 2020), isoprene (Chen et al., 2020b), limonene (Cai et al., 2020), and limonene (Boris et al., 2016), respectively.

**References:**

Chen, Y., et al. (2020). "Heterogeneous Hydroxyl Radical Oxidation of Isoprene-Epoxydiol-Derived Methyltetrol Sulfates: Plausible Formation Mechanisms of Previously Unexplained Organosulfates in Ambient Fine Aerosols." Environmental Science & Technology Letters 7(7): 460-468.

Zhang, Y., et al. (2019). "Joint Impacts of Acidity and Viscosity on the Formation of

Secondary Organic Aerosol from Isoprene Epoxydiols (IEPOX) in Phase Separated Particles." ACS Earth and Space Chemistry 3(12): 2646-2658.

**References**

Apsokardu, M. J., and Johnston, M. V.: Nanoparticle growth by particle-phase chemistry, Atmos. Chem. Phys., 18, 1895-1907, 10.5194/acp-18-1895-2018, 2018.

Boris, A. J., Lee, T., Park, T., Choi, J., Seo, S. J., and Collett Jr, J. L.: Fog composition at Baengnyeong Island in the eastern Yellow Sea: detecting markers of aqueous atmospheric oxidations, Atmos. Chem. Phys., 16, 437-453, 10.5194/acp-16-437-2016,
2016.

Cai, D., Wang, X., Chen, J., and Li, X.: Molecular characterization of organosulfates in highly polluted atmosphere using ultra-high-resolution mass spectrometry, J. Geophys. Res.-Atmos., 125, 10.1029/2019jd032253, 2020.

Charan, S. M., Buenconsejo, R. S., and Seinfeld, J. H.: Secondary organic aerosol yields
from the oxidation of benzyl alcohol, Atmos. Chem. Phys., 20, 13167-13190, 10.5194/acp-20-13167-2020, 2020.

Chen, L., Bao, Z., Wu, X., Li, K., Han, L., Zhao, X., Zhang, X., Wang, Z., Azzi, M., and Cen, K.: The effects of humidity and ammonia on the chemical composition of secondary aerosols from toluene/NOx photo-oxidation, Sci. Total Environ., 728,
138671, 10.1016/j.scitotenv.2020.138671, 2020a.

Chen, T., Liu, Y., Ma, Q., Chu, B., Zhang, P., Liu, C., Liu, J., and He, H.: Significant source of secondary aerosol: formation from gasoline evaporative emissions in the presence of $SO_2$ and $NH_3$, Atmos. Chem. Phys., 19, 8063-8081, 10.5194/acp-19-8063-2019, 2019.

Chen, Y., Zhang, Y., Lambe, A. T., Xu, R., Lei, Z., Olson, N. E., Zhang, Z., Szalkowski, T., Cui, T., Vizuete, W., Gold, A., Turpin, B. J., Ault, A. P., Chan, M. N., and Surratt, J. D.: Heterogeneous Hydroxyl Radical Oxidation of Isoprene-Epoxydiol-Derived Methyltetrol Sulfates: Plausible Formation Mechanisms of Previously Unexplained Organosulfates in Ambient Fine Aerosols, Environ. Sci. Technol. Lett., 7, 460-468,
10.1021/acs.estlett.0c00276, 2020b.

Chu, B., Zhang, X., Liu, Y., He, H., Sun, Y., Jiang, J., Li, J., and Hao, J.: Synergetic formation of secondary inorganic and organic aerosol: effect of $SO_2$ and $NH_3$ on particle formation and growth, Atmos. Chem. Phys., 16, 14219-14230, 2016.

Han, Y., Gong, Z., Liu, P., de Sá, S. S., McKinney, K. A., and Martin, S. T.: Influence
of Particle Surface Area Concentration on the Production of Organic Particulate Matter in a Continuously Mixed Flow Reactor, Environ. Sci. Technol., 53, 4968-4976, 10.1021/acs.est.8b07302, 2019.

Jaoui, M., Edney, E. O., Kleindienst, T. E., Lewandowski, M., Offenberg, J. H., Surratt, J. D., and Seinfeld, J. H.: Formation of secondary organic aerosol from irradiatedα-
pinene/toluene/NOxmixtures and the effect of isoprene and sulfur dioxide, J. Geophys. Res., 113, 10.1029/2007jd009426, 2008.

Jiang, X., Chen, L., You, B., Liu, Z., Wang, X., and Du, L.: Joint impacts of atmospheric SO2 and NH3 on the formation of nanoparticles from photooxidation of a typical biomass burning compound, Environmental Science: Nano, 10.1039/d0en00520g,
2020.

Kleindienst, T. E., Edney, E. O., Lewandowski, M., Offenberg, J. H., and Jaoui, M.:

Secondary organic carbon and aerosol yields from the irradiations of isoprene and α-pinene in the presence of NO$_x$ and SO$_2$, Environ. Sci. Technol., 40, 3807-3812, 10.1021/es052446r, 2006.

Liu, C., Chen, T., Liu, Y., Liu, J., He, H., and Zhang, P.: Enhancement of secondary organic aerosol formation and its oxidation state by SO2 during photooxidation of 2-methoxyphenol, Atmos. Chem. Phys., 19, 2687-2700, 10.5194/acp-19-2687-2019, 2019.

Liu, T., Wang, X., Hu, Q., Deng, W., Zhang, Y., Ding, X., Fu, X., Bernard, F., Zhang, 395 Z., Lu, S., He, Q., Bi, X., Chen, J., Sun, Y., Yu, J., Peng, P., Sheng, G., and Fu, J.: Formation of secondary aerosols from gasoline vehicle exhaust when mixing with SO$_2$, Atmos. Chem. Phys., 16, 675-689, 10.5194/acp-16-675-2016, 2016.

McVay, R. C., Cappa, C. D., and Seinfeld, J. H.: Vapor–Wall Deposition in Chambers: Theoretical Considerations, Environ. Sci. Technol., 48, 10251-10258, 400 10.1021/es502170j, 2014.

Nah, T., McVay, R. C., Zhang, X., Boyd, C. M., Seinfeld, J. H., and Ng, N. L.: Influence of seed aerosol surface area and oxidation rate on vapor wall deposition and SOA mass yields: a case study with α-pinene ozonolysis, Atmos. Chem. Phys., 16, 9361-9379, 10.5194/acp-16-9361-2016, 2016.

Riva, M., Chen, Y., Zhang, Y., Lei, Z., Olson, N. E., Boyer, H. C., Narayan, S., Yee, L. D., Green, H. S., Cui, T., Zhang, Z., Baumann, K., Fort, M., Edgerton, E., Budisulistiorini, S. H., Rose, C. A., Ribeiro, I. O., RL, E. O., Dos Santos, E. O., Machado, C. M. D., Szopa, S., Zhao, Y., Alves, E. G., de Sa, S. S., Hu, W., Knipping, E. M., Shaw, S. L., Duvoisin Junior, S., de Souza, R. A. F., Palm, B. B., Jimenez, J. L., 410 Glasius, M., Goldstein, A. H., Pye, H. O. T., Gold, A., Turpin, B. J., Vizuete, W., Martin, S. T., Thornton, J. A., Dutcher, C. S., Ault, A. P., and Surratt, J. D.: Increasing Isoprene Epoxydiol-to-Inorganic Sulfate Aerosol Ratio Results in Extensive Conversion of Inorganic Sulfate to Organosulfur Forms: Implications for Aerosol Physicochemical Properties, Environ. Sci. Technol., 53, 8682-8694, 10.1021/acs.est.9b01019, 2019.

Vivanco, M. G., Santiago, M., Martinez-Tarifa, A., Borras, E., Rodenas, M., Garcia-Diego, C., and Sanchez, M.: SOA formation in a photoreactor from a mixture of organic gases and HONO for different experimental conditions, Atmos. Environ., 45, 708-715, 10.1016/j.atmosenv.2010.09.059, 2011.

Wyche, K. P., Monks, P. S., Ellis, A. M., Cordell, R. L., Parker, A. E., Whyte, C., 420 Metzger, A., Dommen, J., Duplissy, J., Prevot, A. S. H., Baltensperger, U., Rickard, A. R., and Wulfert, F.: Gas phase precursors to anthropogenic secondary organic aerosol: detailed observations of 1,3,5-trimethylbenzene photooxidation, Atmos. Chem. Phys., 9, 635-665, 10.5194/acp-9-635-2009, 2009.

Ye, J., Abbatt, J. P. D., and Chan, A. W. H.: Novel pathway of SO$_2$ oxidation in the 425 atmosphere: reactions with monoterpene ozonolysis intermediates and secondary organic aerosol, Atmos. Chem. Phys., 18, 5549-5565, 10.5194/acp-18-5549-2018, 2018.

Zhang, R., Wang, G., Guo, S., Zamora, M. L., Ying, Q., Lin, Y., Wang, W., Hu, M., and Wang, Y.: Formation of urban fine particulate matter, Chem. Rev., 115, 3803-3855, 10.1021/acs.chemrev.5b00067, 2015.

Zhang, X., Cappa, C. D., Jathar, S. H., McVay, R. C., Ensber, J. J., Kleeman, M. J., and

Seinfeld, J. H.: Influence of vapor wall loss in laboratory chambers on yields of secondary organic aerosol, P. Natl. Acad. Sci. USA, 111, 5802-5807, 2014.

Zhang, Y., Chen, Y., Lei, Z., Olson, N. E., Riva, M., Koss, A. R., Zhang, Z., Gold, A., Jayne, J. T., Worsnop, D. R., Onasch, T. B., Kroll, J. H., Turpin, B. J., Ault, A. P., and

Surratt, J. D.: Joint Impacts of Acidity and Viscosity on the Formation of Secondary Organic Aerosol from Isoprene Epoxydiols (IEPOX) in Phase Separated Particles, ACS Earth Space Chem., 3, 2646-2658, 10.1021/acsearthspacechem.9b00209, 2019.

Zhao, D., Schmitt, S. H., Wang, M., Acir, I.-H., Tillmann, R., Tan, Z., Novelli, A., Fuchs, H., Pullinen, I., Wegener, R., Rohrer, F., Wildt, J., Kiendler-Scharr, A., Wahner, A., and

Mentel, T. F.: Effects of $NO_x$ and $SO_2$ on the secondary organic aerosol formation from photooxidation of α-pinene and limonene, Atmos. Chem. Phys., 18, 1611-1628, 10.5194/acp-18-1611-2018, 2018.